# Effects of Nitric Oxide on Bladder Detrusor Overactivity through the NRF2 and HIF-1α Pathways: A Rat Model Induced by Metabolic Syndrome and Ovarian Hormone Deficiency

**DOI:** 10.3390/ijms252011103

**Published:** 2024-10-16

**Authors:** Hung-Yu Lin, Jian-He Lu, Rong-Jyh Lin, Kuang-Shun Chueh, Tai-Jui Juan, Jing-Wen Mao, Yi-Chen Lee, Shu-Mien Chuang, Mei-Chen Shen, Ting-Wei Sun, Yung-Shun Juan

**Affiliations:** 1School of Medicine, College of Medicine, I-Shou University, Kaohsiung 84001, Taiwan; ed100464@edah.org.tw; 2Division of Urology, Department of Surgery, E-Da Cancer Hospital, I-Shou University, Kaohsiung 824005, Taiwan; 3Division of Urology, Department of Surgery, E-Da Hospital, I-Shou University, Kaohsiung 82445, Taiwan; 4Center for Agricultural, Forestry, Fishery, Livestock and Aquaculture Carbon Emission Inventory and Emerging Compounds, General Research Service Center, National Pingtung University of Science and Technology, Pingtung County 912301, Taiwan; toddherpuma@mail.npust.edu.tw; 5Department of Parasitology, School of Medicine, College of Medicine, Kaohsiung Medical University, Kaohsiung 807378, Taiwan; rjlin@kmu.edu.tw; 6Graduate Institute of Clinical Medicine, College of Medicine, Kaohsiung Medical University, Kaohsiung 807378, Taiwan; spacejason69@yahoo.com.tw; 7Department of Urology, Kaohsiung Municipal Ta-Tung Hospital, Kaohsiung 80661, Taiwan; 8Department of Urology, Kaohsiung Medical University Hospital, Kaohsiung 80756, Taiwan; blast2337@gmail.com (J.-W.M.); u9181002@gmail.com (S.-M.C.); bear5824@gmail.com (M.-C.S.); ting.wei0220@gmail.com (T.-W.S.); 9Kaohsiung Armed Forces General Hospital, Kaohsiung 802301, Taiwan; terryjuan@gmail.com; 10Department of Thoracic Surgery Division, Kaohsiung Veterans General Hospital, Kaohsiung 813414, Taiwan; 11Department of Anatomy, School of Medicine, College of Medicine, Kaohsiung Medical University, Kaohsiung 807378, Taiwan; yichen83@kmu.edu.tw; 12Department of Urology, College of Medicine, Kaohsiung Medical University, Kaohsiung 807378, Taiwan

**Keywords:** nitric oxide, metabolic syndrome, ovarian hormone deficiency, overactive bladder, high-fat high-sugar diet

## Abstract

Metabolic syndrome (MetS) includes cardiovascular risk factors like obesity, dyslipidemia, hypertension, and glucose intolerance, which increase the risk of overactive bladder (OAB), characterized by urgency, frequency, urge incontinence, and nocturia. Both MetS and ovarian hormone deficiency (OHD) are linked to bladder overactivity. Nitric oxide (NO) is known to reduce inflammation and promote healing but its effect on bladder overactivity in MetS and OHD is unclear. This study aimed to investigate NO’s impact on detrusor muscle hyperactivity in rats with MetS and OHD. Female Sprague-Dawley rats were divided into seven groups based on diet and treatments involving L-arginine (NO precursor) and L-NAME (NOS inhibitor). After 12 months on a high-fat, high-sugar diet with or without OVX, a cystometrogram and tracing analysis of voiding behavior were used to identify the symptoms of detrusor hyperactivity. The MetS with or without OHD group had a worse bladder contractile response while L-arginine ameliorated bladder contractile function. In summary, MetS with or without OHD decreased NO production, reduced angiogenesis, and enhanced oxidative stress to cause bladder overactivity, mediated through the NF-kB signaling pathway, whereas L-arginine ameliorated the symptoms of detrusor overactivity and lessened oxidative damage via the NRF2/HIF-1α signaling pathway in MetS with or without OHD-induced OAB.

## 1. Introduction

The World Health Organization (WHO) defines the essential components of metabolic syndrome (MetS) as having obesity, dyslipidemia, hypertension, and glucose intolerance [1]. In addition to cardiovascular disease, hypertension, diabetes, and obesity, studies have revealed that MetS is also associated with bladder voiding dysfunction and erectile dysfunction. People with MetS have an increased risk of overactive bladder (OAB), which deteriorates bladder storage function [2,3]. According to the International Continence Society definition, OAB is characterized by a set of urinary symptoms: “urgency, with or without urge incontinence, usually with frequency and nocturia” [4]. The mechanism for OAB may involve muscarinic receptors, bladder innervation, purinergic receptors, and an abnormal increase in cytokines [5,6]. For instance, the overexpression of urothelial transient receptor potential vanilloid 1 (TRPV1) [7] and P2X3 receptors [8], along with the C-fiber hypersensitivity, is linked to urgency and detrusor overactivity (DO) in humans [9]. For the relationship between MetS and OAB, studies showed that men with MetS had increased odds of lower urinary tract symptoms (LUTSs), supporting the role of metabolic disturbance in the etiology of LUTSs [10]. Several inflammatory markers, such as IL-1, IL-6 [11,12], NF-κB [13], tumor necrosis factor (TNF-α) [14,15], C-reactive protein [16,17], adiponectin [18], and fibronectin, are associated with the MetS. Patients with MetS commonly experience central obesity, where visceral adipose tissue plays a significant role by producing inflammatory cytokines that contribute to both inflammation and endothelial dysfunction.

Diets with high cholesterol may lead to the development of MetS and OAB [19,20]. Several animal models have been established to study the symptoms and signs of MetS. Animal studies have shown that administering a fructose-rich diet or a high-fat, high-sugar (HFHS) diet to rats induces MetS [21,22,23]. However, most of these abnormalities can be reversed by withdrawing from the HFHS diet [24]. In a fructose-fed rat model, 62.5% of male rats exhibited unstable bladder contractions compared to the control group. Moreover, increased expression of M2- and M3-muscarinic receptor mRNA and protein levels has been linked to detrusor overactivity [25]. In addition to OAB, feeding rats with a fructose-rich diet may also develop acontractile detrusor [26]. In animal models, a hypercholesterol diet fed to rats can induce hypertrophy of bladder detrusor cells, reduce bladder capacity, and produce symptoms/signs of OAB. In fructose-fed rats with induced detrusor overactivity, there was a disruption in smoothelin regulation and a reduction in Bcl-2 expression, leading to an increase in apoptotic cells in the bladder wall. M2- and M3-muscarinic receptors and P2X1 receptors were up-regulated in the fructose-fed rats with detrusor overactivity [26]. These data suggested that the MetS is closely related to OAB. However, the exact mechanism of MetS-associated bladder dysfunction remains unclear.

Ovarian hormone deficiency (OHD) is associated with an increased risk of MetS [27]. Postmenopausal women with OHD exhibit various urinary dysfunctions, including OAB symptoms, stress incontinence, and recurrent urinary tract infections [28,29]. It is estimated that up to 40% of postmenopausal women suffer from symptomatic urogenital atrophy [30]. Experimentally, the physiological condition of OHD or postmenopausal status is simulated in rat models treated with bilateral ovariectomy (OVX), inducing symptoms of detrusor hyperactivity [31,32,33]. Ovariectomized rats exhibited urinary dysfunction, such as increased post-voiding residual (PVR) urine, decreased voiding efficiency, detrusor hyperactivity, and altered coordination between the urethral sphincter and bladder detrusor [34]. In the ER-ß^−/−^ female mice, pathological features, such as urothelial ulceration, atrophy, and bladder hyperactivity, were consistent with human interstitial cystitis and bladder pain syndrome (IC/BPS) [35]. In ovariectomized rat studies, estrogen deficiency may diminish bladder blood flow and tissue hypoxia, leading to OAB symptoms, stress urinary incontinence, and recurrent urinary tract infections. OHD in OVX rats appeared as bladder hyperactivity [32]. Our earlier study indicated that OVX-induced OHD rats exhibited decreased bladder compliance, increased levels of oxidative damage, interstitial fibrosis, and bladder mucosa apoptosis [36]. OVX-treated rabbits showed notable vascular degeneration and reduced vascular density. However, estradiol treatment induced angiogenic remodeling and enhanced vascular density within the detrusor smooth muscle bundles, thereby improving OAB symptoms [37,38]. On the other hand, estrogen receptors α and ß in rat models are expressed in bladder afferent neurons in the lumbosacral dorsal root ganglia [39]. Both estrogen α and ß receptors are expressed in neurons co-stained with a capsaicin-sensitive, nociceptive ion channel, transient receptor potential vanilloid 1 (TRPV1) [40]. In rat nociceptor neurons, 17β-estradiol plays a role in activating estrogen receptor signaling and inhibiting capsaicin-induced activation of TRPV1, thereby regulating bladder pain [41]. As a result, estrogen may diminish the stimulatory effects of capsaicin and regulate bladder pain. Although the pathophysiological changes in the bladder of OVX animals have been reported, the mechanism underlying bladder dysfunction remains unclear.

Nitric oxide (NO) acts as a source of free radicals and plays an important intercellular signaling messenger in pathological processes. NO is synthesized by nitric oxide synthase (NOS), which uses L-arginine as a substrate to synthesize NO and L-citrulline. There are three isoforms of NOS: neuronal NOS (nNOS), inducible NOS (iNOS), and endothelial NOS (eNOS). NOS could be stimulated by L-arginine and inhibited by a competitive inhibitor of NOS, NG-nitro-L-arginine methyl ester (L-NAME). Administration with NOS substrate might prevent hypercholesterolemic microvascular inflammation, attenuate the number of endothelial apoptotic cells, and reverse endothelial dysfunction. These studies revealed that NO plays an important role in MetS. Estradiol-induced vascular relaxation may be enhanced through eNOS activity and could be counteracted by L-NAME [42]. Moreover, treatment with L-NAME obviously reduced the nitrotyrosine generation [43]. A deficiency in endothelial-derived NO is considered the main defect linking insulin resistance and endothelial dysfunction. Moreover, endothelial dysfunction and insulin resistance are consistently associated with MetS. Treatment with L-NAME can reduce apoptosis induced by ischemia-reperfusion (I/R) injury in the bladder and significantly increase the contractile responses compared with the ischemia-reperfusion (I/R) group without L-NAME [44,45,46].

NO was thought to have a dual role, serving as a regulator under physiological and pathophysiological conditions. NO is a potent vascular regulator that controls blood flow in both the LUT and the cavernous smooth muscle [47]. Urinary frequency significantly increased in animals with pelvic ischemia [48]. NO modulates bladder neck and urethra muscle relaxation, plays an important role in prostatic cell growth, and influences the reproductive system’s neuronal function. NO may have varying roles in the pathophysiology of bladder dysfunction, especially in the MetS with or without OHD-induced OAB.

The present study used OVX-treated rats to mimic the postmenopausal status to induce detrusor hyperactivity. The role of NO in the MetS with or without OHD-induced bladder dysfunction is still not fully understood. It was hypothesized that NO has a therapeutic effect by increasing bladder angiogenesis and neurogenesis and decreasing oxidative stress, thereby improving detrusor overactivity via nuclear factor erythroid 2-related factor 2 (NRF2) and hypoxia-inducible factor-1α (HIF-1α) activation in a rat model with MetS with or without OHD. Thus, this study aimed to evaluate the effect of NO on bladder dysfunction after a HFHS diet with or without OVX-induced OAB in the presence of NO precursor (L-arginine) and/or NOS inhibitor (L-NAME). The obtained information will be valuable in providing new insights into a better understanding of the effect of NO on the MetS and postmenopausal status with OAB people. It was found that administering L-arginine to partial bladder outlet obstruction (PBOO) rabbits can increase bladder compliance and reverse bladder dysfunction [49]. In a pelvic ischemia model, created by inducing pelvic arterial atherosclerosis, bladder ischemia up-regulated the expression of hypoxia-inducible factor-1α (HIF-α), transforming growth factor-b (TGF-β), and vascular endothelial growth factor (VEGF). Also, increased oxidative stress also serves as an important etiologic factor for postmenopausal women suffering from LUTSs. In fructose-fed rats with detrusor overactivity, nitrotyrosine expression was up-regulated in the bladder tissue, showing increased nitrosative stress in fructose-induced MetS and OAB [26]. Moreover, evidence revealed a close link between MetS, chronic inflammation, and oxidative stress.

## 2. Results

### 2.1. Serum Estradiol and Testosterone Concentration Reduced after Bilateral Ovariectomy (OVX)

Rats were fed with a HFHS diet to induce MetS and, bilaterally, OVX surgery was applied to mimic menopausal conditions of OHD to induce bladder overactivity in the rat model. As compared to the control group (44.8 ± 8.5 pg/mL), the serum estradiol concentration was 43.5 ± 7.1 pg/mL for the MetS group, 36.6 ± 5.7 pg/mL for the MetS + L-arginine group, and 38.4 ± 7.4 pg/mL for the MetS + L-NAME group, respectively. There was no significant difference in the serum estradiol concentration among different groups without OVX treatment. However, the serum estradiol concentration was significantly decreased four weeks after OVX treatment: 18.9 ± 2.1 pg/mL for the MetS + OVX group (*p* = 0.001), 19.6 ± 3.3 pg/mL for the OVX + MetS + L-arginine group (*p* = 0.001), and 20.3 ± 3.9 pg/mL for the OVX + MetS + L-NAME group (*p* = 0.001) in comparison with the control group.

Moreover, the serum testosterone concentration was significantly decreased four weeks after OVX treatment: 35.0 ± 4.7 pg/mL for the MetS + OVX group (*p* = 0.001), 36.4 ± 9.3 pg/mL for the OVX + MetS + L-arginine group (*p* = 0.001), and 33.2 ± 4.7 pg/mL for the OVX + MetS + L-NAME group (*p* = 0.001) in comparison with the control group (61.2 ± 10.9 pg/mL), the MetS group (67.2 ± 13.6 pg/mL), the MetS + L-arginine group (64.8 ± 12.2 pg/mL), and the MetS + L-NAME group (61.4 ± 9.9 pg/mL), respectively (Table 1). Table 1 shows OVX significantly decreased serum estradiol (F (6, 98) = 59.4048, *p* < 0.0001) and testosterone (F (6, 98) = 37.0431, *p* < 0.0001) concentration in comparison with the control group and MetS group without OVX. The results revealed that serum estradiol and testosterone deficiency were induced by bilateral OVX surgery.

### 2.2. Physical Characteristics

After 12 months of OVX, physical characteristics are shown to be present in Table 1, including water intake, urine output, body weight, bladder weight, the ratio of bladder weight to body weight, waist circumference, systolic pressure, diastolic pressure, and mean arterial pressure (MAP). The different groups showed no significant difference in water intake and urine output. The physical indicators of MetS developed significant increases in body weight, the ratio of bladder weight to body weight, waist circumference, and blood pressure (systolic pressure, diastolic pressure, and MAP). Results indicated that long-term HFHS feeding with or without OHD caused a profound negative effect on the level of body weight, bladder weight, the ratio of bladder weight to body weight, and blood pressure, resulting in bladder pathological alteration.

### 2.3. Changes in Liver Morphology and Serum Indicators

MetS increases the risk of liver dysfunction. The changes in liver morphology and physical indicators after 12 months of standard diet feeding (control group, Figure 1A) and HFHS diet feeding without OVX in the MetS group (Figure 1B), the MetS + L-arginine group (Figure 1C), and the MetS + L-NAME group (Figure 1D) are shown in Figure 1, as are HFHS diet feeding with OVX in the MetS + OVX group (Figure 1E), the MetS + OVX + L-arginine group (Figure 1F), and the MetS + OVX + L-NAME group (Figure 1G). In the control group (Figure 1A), the liver showed a dark-red appearance. However, the liver photography after MetS with or without OVX showed a fatty and swollen appearance (Figure 1B–G). Particularly, liver appearance in the MetS + OVX group (Figure 1E) and the MetS + OVX + L-NAME group (Figure 1G) was obviously fatty and edematous. However, the appearance of the MetS + L-arginine group liver appeared to be improving compared to the MetS group.

The characteristics of MetS by serum parameters developed, including glutamate oxaloacetate transaminase (GOT), glutamate pyruvate transaminase (GPT), triglycerides, cholesterol, low-density lipoprotein (LDL), high-density lipoprotein (HDL), glucose, creatinine, insulin, lactate dehydrogenase (LDH), and alkaline phosphatase (ALK-P) (Figure 1H). As compared to the control group, serum parameters, including GOT, GPT, triglycerides, cholesterol, LDL, HDL, glucose, creatinine, LDH, and ALK-P, were significantly elevated in the MetS group, the MetS + L-arginine group, the MetS + L-NAME group, the MetS + OVX group, the MetS + OVX + L-arginine group, and the MetS + OVX + L-NAME group. However, there is no difference in insulin analysis between the control group and the MetS with or without OHD groups (Table 1). Moreover, the level of LDH and Alk-P was elevated in the groups as compared to the control group, with significant decreases in the ratio of Alk-P to LDH.

These results suggest that the combination of MetS with OHD had a severe negative impact on biochemical parameters and led to significant deterioration in renal function. However, these adverse effects on physical indicators and biochemical parameters were reversed in the MetS + L-arginine group compared to the MetS group. (Table 1). However, there was a limitation on the L-arginine treatment for the MetS + L-arginine group and the MetS + OVX + L-arginine group to restore the control level.

### 2.4. L-Arginine Treatment Ameliorated Bladder Hyperactivity

The recordings of the 1 h cystometry (CMG) and 24 h micturition frequency were shown to examine the bladder function after MetS with or without OHD, including peak micturition pressure (arrows), micturition frequency, micturition interval, voided volume, and non-voided contraction (star), in Table 1 and Figure 2. The CMG data of the MetS with or without OHD groups significantly increased intravesical pressure (Figure 2A,C) and micturition frequency (Figure 2A,B,D) compared to the control group but decreased voiding volume (Figure 2B,E). Specifically, the MetS + OVX group and the MetS + OVX + L-NAME group exhibited bladder overactivity, characterized by increased peak micturition pressure (indicated by arrows), non-voiding contractions (marked by asterisks), and elevated micturition frequency. Notably, higher bladder pressure suggested reduced bladder compliance.

By contrast, after L-arginine treatment, the bladder capacity significantly increased, whereas peak micturition pressure, micturition frequency, and non-voiding contractions significantly decreased (Figure 2A,C,D). The observations above demonstrated that L-arginine treatment in the MetS + L-arginine and MetS + OVX + L-arginine groups significantly reduced peak micturition pressure, micturition frequency, and non-voiding contractions compared to the MetS and MetS + OVX groups.

From the analysis of micturition behavior, the MetS with or without OHD groups revealed lower voided volume and more micturition frequency than the control group (Figure 2B,D,E). On the contrary, the MetS + L-arginine group and the MetS + OVX + L-arginine group significantly decreased micturition pressure and frequency and increased voided volume as compared with those in the MetS group and the MetS + OVX group. These results suggested that OHD worsened bladder overactivity and led to abnormal detrusor activity in MetS rats. However, L-arginine efficacy in the MetS + L-arginine group and the MetS + OVX + L-arginine group slightly improved but did not fully recover from such deteriorations.

### 2.5. L-Arginine Treatment Improved the Bladder Detrusor Contractile Response

The bladder detrusor contractile response was assessed to evaluate synaptic transmission, receptor activity, and smooth muscle contraction. The results of electrical-field stimulation (EFS), carbachol, and KCl stimulation for contractile responses on bladder strips are shown in Figure 3. The bladder strips in the MetS group, the MetS + L-NAME group, the MetS + OVX group, and the MetS + OVX +L-NAME group had higher contractile responses induced by EFS at 2, 8, and 32 Hz compared with the control group; the MetS + L-arginine and the MetS + OVX + L-arginine groups exhibited significantly lower contractile responses compared to the MetS and MetS + OVX groups (Figure 3A,D). Similar results were obtained for muscle strip stimulation induced by carbachol (Figure 3B,D) and KCl (Figure 3C,D).

Therefore, L-arginine treatment significantly decreased the detrusor contractile response to various forms of stimulation in the MetS + L-arginine group and the MetS + OVX + L-arginine group. These results suggest that the MetS group, with or without OHD, exhibited a worse bladder contractile response, leading to bladder contractile deficiency, while L-arginine improved bladder contractile function.

### 2.6. L-Arginine Improved Bladder Pathological Alteration in Association with MetS and OHD

Masson’s trichrome stain was shown to evaluate the pathological changes in the bladder following MetS with or without OHD in Figure 4. In the control group (Figure 4A,A’), there were three to five layers of urothelium (UL), with only sparse collagen distributed in the suburothelium (SL) and the muscular layer (ML). In contrast, the MetS group exhibited significant interstitial fibrosis and collagen accumulation (arrows) between detrusor smooth muscle (DSM) bundles (Figure 4B,B’), the MetS + L-NAME group (Figure 4D,D’), the MetS + OVX group (Figure 4E,E’), and the MetS + OVX + L-NAME group (Figure 4G,G’). Moreover, morphological evaluation of the MetS + OVX group (Figure 4E,E’) revealed denuded urothelial mucosa, along with defective and thinning urothelium (black arrows) and significant interstitial fibrosis (green arrows) when compared to the MetS group (Figure 4B,B’). However, bladder tissues from the MetS + L-arginine and MetS + OVX + L-arginine groups showed a significant reduction in interstitial fibrosis and collagen accumulation compared to the MetS and MetS + OVX groups.

The expressions of inflammatory and fibrosis markers (TGF-β1, fibronectin, and type I collagen) were analyzed using Western Blotting (Figure 4H,I). Compared to the control, the levels of TGF-β, fibronectin, and type I collagen proteins were significantly elevated in the MetS group compared to the MetS + L-arginine group, as well as in the MetS + OVX group compared to the MetS + OVX + L-arginine group (TGF-β: F (6, 35) = 19.1265, *p* < 0.0001; Fibronectin: F (6, 35) = 21.4826, *p* < 0.0001; Collagen I: F (6, 35) = 20.7045, *p* < 0.0001). The expressions of inflammatory and fibrosis markers were significantly enhanced in the MetS group, the MetS + L-NAME group, the MetS + OVX group, and the MetS + OVX + L-NAME group, indicating an increase in bladder interstitial fibrosis, mucosal damage, and reduced bladder compliance as compared to the control group. After L-arginine treatment in the MetS + L-arginine group and the MetS + OVX + L-arginine group, these expressions were significantly decreased as compared to the MetS group and the MetS + OVX group. Nevertheless, the administration of L-arginine alleviated bladder interstitial fibrosis between DSM bundles, improved mucosal damage, and micturated compliance.

### 2.7. L-Arginine Improved Bladder Urothelial Regeneration and Interstitial Cell (IC) Generation

In the control group (Figure 5A), the urothelial layer (UL; yellow arrows), analyzed by E-Cadherin staining, had three to five layers. In contrast, the bladders in the MetS groups, with or without OHD, displayed a thinner and defective urothelial mucosa in the UL (black arrows). In particular, defective urothelial mucosa was significantly enhanced in the MetS group, the MetS + L-NAME group, the MetS + OVX group, and the MetS + OVX + L-NAME group as compared with the control group. However, morphological evaluation of the MetS + L-arginine group (Figure 5C) and the MetS + OVX + L-arginine group (Figure 5F) showed an increased thicker layer of UL to improve bladder damage. However, there was a limitation on the L-arginine treatment for the MetS + L-arginine group and the MetS + OVX + L-arginine group to restore the control level.

Western Blot analysis of E-Cadherin, CK14, C-Kit, vimentin, and PDGFR expressions was investigated (Figure 5H,I). All protein expressions in the MetS group, the MetS + L-NAME group, the MetS + OVX group, and the MetS + OVX + L-NAME group were significantly reduced compared to the control group (E-Cadherin: F (6, 35) = 49.9328, *p* < 0.0001; CK-14: F (6, 35) = 13.2992, *p* < 0.0001; C-kit: F (6, 35) = 17.0987, *p* < 0.0001; PDGFR: F (6, 35) = 12.3221, *p* < 0.0001; Vimentin: F (6, 35) = 31.5924, *p* < 0.0001). In contrast, expressions in the MetS + L-arginine group and the MetS + OVX + L-arginine group were significantly elevated compared to the MetS and MetS + OVX groups. Additionally, protein levels were noticeably increased in the MetS + L-NAME group and the MetS + OVX + L-NAME group compared to the MetS and MetS + OVX groups (Figure 5H,I). Such that the levels of E-Cadherin, CK14, C-kit, and PDGFR were increased in the MetS + L-arginine group and the MetS + OVX + L-arginine group to an extent similar to the control group. According to the above data, L-arginine treatment stimulated CK14+ cell proliferation in the urothelial basal layer and improved interstitial vimentin+ myofibroblast proliferation to improve mucosal regeneration and alter bladder remodeling for bladder repair in the pathogenesis of OHD-induced OAB.

### 2.8. L-Arginine Altered Bladder Angiogenic Remodeling

To elucidate whether L-arginine (NO precursor) treatment improved bladder angiogenesis in a rat model of MetS with or without OHD-induced detrusor hyperactivity, angiogenesis-related markers (alpha smooth muscle actin (α-SMA), Laminin, and VEGF) were quantified by immunostaining (Figure 6A–G) and Western Blots (Figure 6H,I). The myofibroblastic phenotype was assessed through immunostaining and Western Blot analysis of α-SMA and Laminin expression. In the control group (Figure 6A), α-SMA immunostaining (yellow arrows) was widely distributed in myofibroblasts and the smooth muscle of microvessels located beneath the urothelial basal layer, within the SL and ML, respectively. The α-SMA staining was reduced in the bladder tissues of the MetS group (Figure 6B) and the MetS + OVX group (Figure 6E). In contrast, α-SMA immunostaining was prominently expressed in the myofibroblasts, microvessels, and vessels of the urothelial basal layer, SL, and ML in the MetS + L-arginine group (Figure 6C), the MetS + L-NAME group (Figure 6D), the MetS + OVX + L-arginine group (Figure 6F), and the MetS + OVX + L-NAME group (Figure 6G), compared to the MetS and MetS + OVX groups. Additionally, the protein levels were obviously enhanced in the MetS + L-NAME group and the MetS + OVX + L-NAME group compared to the MetS group and the MetS + OVX group (Figure 6H,I).

Western Blot analysis further investigated the protein levels of α-SMA, Laminin, and VEGF (Figure 6H,I). The expression of α-SMA, Laminin, and VEGF, as shown in the MetS group and the MetS + OVX group, significantly declined as compared with the control group (α-SMA: F (6, 35) = 22.7660, *p* < 0.0001; Laminin: F (6, 35) =11.1900, *p* < 0.0001; VEGF: F (6, 35) = 49.9328, *p* < 0.0001). After L-arginine treatment, these expressions were significantly enhanced, as shown in the MetS + L-arginine group as compared to the MetS group, as well as the MetS + OVX + L-arginine group as compared to the MetS + OVX group. Additionally, the protein levels were meaningfully enhanced in the MetS + L-arginine group and the MetS + OVX + L-arginine group compared to the MetS + L-NAME group and the MetS + OVX + L-NAME group. According to the above data, L-arginine stimulated α-SMA+ and Laminin+ associated with myofibroblasts in the SL and ML to increase angiogenic remodeling, which regulated blood flow for bladder repair in the pathogenesis of MetS with or without OHD-induced detrusor overactivity.

### 2.9. The Effect of L-Arginine Promoted Bladder Neurogenesis

To examine the effect of L-arginine (a NO precursor) on bladder neurogenesis, including synaptic transmission, neuronal regeneration, and receptor response, the expression of neuronal markers (neurofilament, glial fibrillary acidic protein (GFAP)), neuronal nuclei (NeuN), muscarinic receptors (M2 and M3), and purinergic receptors (P2X3 and P2X7) were analyzed through immunostaining and Western Blotting (Figure 7). In the control group (Figure 7A), neurofilament staining (yellow arrows) and neural ganglia (green arrows) were primarily expressed in the submucosal layer (SL, lamina propria) and muscular layer (ML). In comparison with the control group, less neurofilament staining (yellow arrows) and neural ganglion (green arrows) were restricted to SL and UL of the MetS group (Figure 7B) and the MetS + OVX group (Figure 7E). However, those stainings were widely distributed in the MetS + L-arginine group (Figure 7C), the MetS + OVX + L-arginine group (Figure 7F), the MetS + L-NAME group (Figure 7D), and the MetS + OVX + L-NAME group (Figure 7G) as compared with the MetS group and the MetS + OVX group. Particularly, the neurofilament labeling (yellow arrows) of the MetS + L-arginine group and the MetS + OVX + L-arginine group was markedly expressed in the SL compared to the MetS + L-NAME group and the MetS + OVX + L-NAME group.

For Western Blot analysis, the markers of neurofilament, NeuN, GFAP, P2X3, and P2X7 were significantly suppressed in the MetS group and the MetS + OVX group as compared with the control group, whereas the expressions of M2 and M3 were meaningfully increased (Neurofilament: F (6, 35) = 23.4399, *p* < 0.0001; NeuN: F (6, 35) = 9.4984, *p* < 0.0001; GFAP: F (6, 35) = 22.6599, *p* < 0.0001; M2: F (6, 35) =29.3770, *p* < 0.0001; M3: F (6, 35) = 22.3518, *p* < 0.0001; P2X3: F (6, 35) = 39.8812, *p* < 0.0001; P2X7: F (6, 35) = 81.2329, *p* < 0.0001). Additionally, the markers of neurofilament, NeuN, GFAP, P2X3, and P2X7 were obviously increased in the MetS + L-arginine group, the MetS + OVX + L-arginine group, the MetS + L-NAME group, and the MetS + OVX + L-NAME group as compared with the MetS group and the MetS + OVX group (Figure 7I). Especially, the levels of the MetS + L-arginine group and the MetS + OVX + L-arginine group were markedly expressed compared to the MetS + L-NAME group and the MetS + OVX + L-NAME group. Based on the immunostaining and Western Blot results, MetS with or without OHD led to neuronal degeneration. However, L-arginine was shown to enhance neurogenesis and receptor response, thereby improving bladder overactivity in a rat model of MetS-induced detrusor overactivity, with or without OHD.

### 2.10. The Effect of L-Arginine Increased NO Production and Decreased Oxidative Stress

To determine the role of L-arginine on oxidative stress in MetS and OHD MetS with or without OHD-induced detrusor overactivity, the expressions of NOS (iNOS, eNOS, and nNOS), transcription factors (HIF-1α, NRF2, and NFκB), and oxidative stress markers (DNP and nitrotyrosine) were measured by Western Blots (Figure 8). The expressions of all NOS, HIF-1α, and NRF2 were significantly suppressed in the MetS group and the MetS + OVX group in comparison with the control group, whereas the expressions of NFκB-P65, DNP, and nitrotyrosine were significantly overexpressed (iNOS: F (6, 35) = 47.8491, *p* < 0.0001; eNOS: F (6, 35) = 43.3941, *p* < 0.0001; nNOS: F (6, 35) = 9.1644, *p* < 0.0001; HIF-1α: F (6, 35) =50.2806, *p* < 0.0001; NRF2: F (6, 35) = 44.9870, *p* < 0.0001; NFκB-p65: F (6, 35) = 27.1131, *p* < 0.0001; DNP: F (6, 35) =70.1759, *p* < 0.0001; Nitrotyrosine: F (6, 35) = 42.7615, *p* < 0.0001). After L-arginine treatment, the levels of NFκB-P65, DNP, and nitrotyrosine were meaningfully reduced in the MetS + L-arginine group and the MetS + OVX + L-arginine group compared to the MetS group and the MetS + OVX group, whereas the levels of all NOS, HIF-1α, and NRF2 were significantly strengthened. Particularly, the levels of NOS (iNOS, eNOS, and nNOS) and transcription factors (HIF-1α and NRF2) in the MetS + L-arginine group and the MetS + OVX + L-arginine group were markedly increased compared to the MetS + L-NAME group and the MetS + OVX + L-NAME group. Therefore, both MetS with or without OHD induced the expressions of oxidative markers in the bladder, whereas L-arginine alleviated the extent of these expressions. These findings demonstrated that MetS with or without OHD status reduced NO production and exacerbated oxidative damage of the bladder, whereas the effect of L-arginine offered a beneficial effect on lessening MetS with or without OHD-related oxidative damages. The above findings demonstrated that MetS with or without OHD enhanced the generation of oxidative stress mediated through the NFκB signaling pathway, whereas L-arginine offered a beneficial effect on lessening MetS and OHD-related oxidative damages via the NRF2/HIF-1α signaling pathway in a rat model of MetS and OHD-induced OAB.

### 2.11. Mitochondria-Elicited Bladder Damage with MetS and OHD

The expressions of mitochondrial respiratory enzyme complexes (NDUFS3, UQCRC1, COX-2, SDHA, and ATPB) were significantly elevated in the MetS group and the MetS + OVX group compared to the control group (NDUFS3: F (6, 35) = 30.4585, *p* < 0.0001; SDHA: F (6, 35) = 12.8988, *p* < 0.0001; UQCRC2: F (6, 35) = 12.5466, *p* < 0.0001; COX-2: F (6, 35) = 15.6375, *p* < 0.0001; ATPB: F (6, 35) = 33.4911, *p* < 0.0001) (Figure 9). After L-arginine treatment, the levels were meaningfully reduced in the MetS + L-arginine group and the MetS + OVX + L-arginine group compared to the MetS group and the MetS + OVX group. Compared with the MetS group and the MetS + OVX group, the levels were decreased in the MetS + L-arginine group and the MetS + OVX + L-arginine group. The above findings suggested a potential rat model for the L-arginine effect on oxidative stress mediated through mitochondria as shown.

The effect of L-arginine increased NO production and decreased oxidative stress. To determine the role of L-arginine in oxidative stress in MetS and OHD MetS with or without OHD-induced detrusor overactivity, the expressions of NOS (iNOS, eNOS, and nNOS), transcription factors (HIF-1α, NRF2, and NFκB), and oxidative stress markers (DNP and nitrotyrosine) were measured by Western Blots (Figure 8). The expressions of all NOS, HIF-1α, and NRF2 were significantly suppressed in the MetS group and the MetS + OVX group in comparison with the control group, whereas the expressions of NFκB-P65, DNP, and nitrotyrosine were significantly overexpressed. After L-arginine treatment, the levels of NFκB-P65, DNP, and nitrotyrosine were meaningfully reduced in the MetS + L-arginine group and the MetS + OVX + L-arginine group compared to the MetS group and the MetS + OVX group, whereas the levels of all NOS, HIF-1α, and NRF2 were significantly strengthened.

The above findings suggest a potential rat model for studying the effects of L-arginine on oxidative stress mediated through mitochondrial pathways. These results indicate that MetS and OHD increased oxidative stress via mitochondrial pathways. Conversely, L-arginine reduced oxidative stress and downregulated mitochondrial signals, suggesting that L-arginine alleviated bladder overactivity through mitochondria-mediated pathways in rats with MetS and OHD. There was a proposed potential mechanism of L-arginine, NO precursor, to improve bladder detrusor hyperactivity in a rat model of OHD. After 12 months of HFHS treated with or without OVX, the characteristics of MetS developed. MetS with or without OHD status showed significant deterioration in bladder capacity and enhanced bladder overactivity. Furthermore, MetS with or without OHD status revealed increased interstitial fibrosis and significant increases in inflammation and fibrosis (Figure 10).

However, rats treated with L-arginine (NO precursor) significantly ameliorated storage dysfunction and the symptoms of MetS with or without OHD-induced OAB. Moreover, the MetS group and the MetS + OVX group resulted in reduced angiogenesis, neurogenesis, inhibited activation of NO, and induced expressions of oxidative markers in the bladder via the NFκB signaling pathway, while L-arginine treatment could reduce the extent of these expressions and improve those via the NRF2/HIF-1α signaling pathway in MetS with or without OHD-induced OAB. According to the present data, the increased expression of mitochondrial respiratory enzyme complexes implied that MetS and OHD may lead to the generation of reactive oxygen species (ROS) while this increase was mitigated by L-arginine treatment.

These results implied that NO may play an important role in regulating bladder function and could regulate the activity of the mitochondria electron transport chain to influence the production of ROS. In summary, MetS with or without OHD enhanced the generation of oxidative stress mediated through the NFκB signaling pathway, whereas L-arginine offered a beneficial effect on lessening oxidative damages via the NRF2/HIF-1α signaling pathway in a rat model of MetS with or without OHD-induced OAB.

## 3. Discussion

The characteristics of MetS developed after 12 months of HFHS feeding treated with or without OVX. There were meaningful increases in body weight, bladder weight, systolic pressure, and MAP, whereas the ratio of bladder weight to body weight was significantly reduced as compared to the control group. The MetS group and the MetS + OVX group showed significant deterioration in bladder capacity and enhancement in bladder overactivity, whereas L-arginine (NO precursor) exhibited significantly ameliorated storage dysfunction. Additionally, rats treated with L-arginine exhibited lower intravesical pressure and better contractile responses to all forms of stimulation compared to those treated with or without L-NAME. Furthermore, the MetS group and the MetS + OVX group revealed increased interstitial fibrosis. Moreover, the MetS group and the MetS + OVX group resulted in reducing angiogenesis and neurogenesis, inhibiting the activation of NO and inducing the expressions of oxidative markers in the bladder, while L-arginine treatment could reduce the extent of these expressions and improve those in MetS with or without OHD-induced OAB. Particularly, the MetS + LNAME group and the MetS + OVX + L-NAME group had higher oxidative stress markers than the MetS + L-arginine group and the MetS + OVX + L-arginine group. In summary, MetS with or without OHD enhanced the generation of oxidative stress mediated through the NFκB signaling pathway, whereas L-arginine offered a beneficial effect on lessening oxidative damages, improving angiogenesis and neurogenesis via the NRF2/HIF1α signaling pathway, as shown in a rat model of MetS with or without OHD-induced OAB.

MetS was diagnosed based on a combination of clinical and laboratory features, such as abdominal obesity, hypertension, hyperglycemia, and dyslipidemia. However, these indicators are symptoms of metabolic disorders, not the underlying causes [50]. Many definitions of MetS have been proposed, with five equal criteria: elevated waist circumference, elevated triglyceride levels, elevated fasting glucose, low HDL levels, and elevated blood pressure [51]. A diagnosis of MetS requires the presence of at least three of these criteria [52]. In this study, after 12 months of HFHS feeding treated with or without OVX, as shown in Table 1 and Figure 1, the characteristics of MetS developed. Our physical indicators revealed that MetS developed significant increases in body weight, bladder weight, the ratio of bladder weight and body weight, waist circumference, systolic pressure, diastolic pressure, and MAP in the MetS group, the MetS + L-arginine group, the MetS + L-NAME group, the MetS + OVX group, the MetS + OVX + L-arginine group, and the MetS + OVX + L-NAME group as compared to the control group. Additionally, liver photography after HFHS feeding with or without OHD showed a fatty, swollen, and edematous appearance (Figure 1A–G). Serum parameters were significantly elevated in the MetS with or without OHD groups as compared to the control group, including GOT, GPT, triglycerides, cholesterol, LDL, glucose, and LDH (except insulin level) (Figure 1 and Table 1). However, there was a limitation on physical indicators and serum parameters for the MetS + L-arginine group and the MetS + OVX + L-arginine group to restore the control level.

In the fructose-fed rat model, male rats on a high-fat diet show increased non-voiding contractions, voiding frequency, and post-voiding pressure. Additionally, MetS accompanied by hyperlipidemia and hyperglycemia leads to excessive ROS production and impaired mitochondrial ATP production [53]. Our previous study revealed that rats fed a HFHS diet for 12 months could develop both MetS and OAB. Moreover, MetS combined with OVX to deprive ovarian-hormone-deteriorated bladder storage dysfunction more profoundly than MetS alone [36]. In this study, our findings suggest that MetS, with or without OHD, led to significant bladder hyperactivity, abnormal detrusor activity, increased micturition frequency, and reduced bladder capacity. However, L-arginine treatment significantly improved bladder capacity and alleviated the symptoms of detrusor overactivity following a HFHS diet, with or without OHD. (Figure 2). Additionally, the MetS with or without OHD groups had worse bladder contractile responses, causing bladder contractile deficiency, while L-arginine ameliorated the bladder contractile function (Figure 3). These results implied that the NO pathway may play an important role in regulating bladder function. However, the effect of MetS in combination with OHD seemed to be more complex than MetS alone.

Menopause was a risk factor for MetS, type 2 diabetes, and cardiovascular diseases. The prevalence of MetS with postmenopause decreased serum estrogen concentration and increased androgen concentration, leading to alterations in body fat distribution and abdominal obesity. Excessive visceral adipose tissue significantly contributed to the synthesis and secretion of bioactive substances, including adipocytokines, proinflammatory cytokines, ROS, prothrombotic factors, and vasoconstrictor agents. Several studies demonstrated that testosterone levels were higher in women with MetS [54]. Furthermore, the ratio of testosterone to estradiol during menopausal status increases the incidence of MetS [55,56]. In this study, OVX significantly decreased the concentration of serum estradiol and testosterone (Table 1). The results showed that serum estradiol and testosterone deficiency was induced by bilateral OVX surgery. Additionally, the physical indicators of MetS with or without OHD developed significant increases in body weight, bladder weight, the ratio of bladder weight to body weight, waist circumference, systolic pressure, diastolic pressure, and MAP as compared to the control group.

NO is a powerful regulator and neurotransmitter responsible for vascular tone for relaxation activity in the LUT [57]. Estradiol-induced vascular relaxation may be enhanced by eNOS activity and can be inhibited by L-NAME (a NOS inhibitor) [42]. In addition, the bladders with OAB decreased NO release from the urothelium in streptozocin-induced diabetic rats [58]. In animal models of myocardial I/R studies, impaired release of NO may be an important factor for I/R injury. Conversely, administration of L-arginine (NO precursor) reduced infarct size and improved postischemic cardiac recovery and endothelial function [59,60,61]. Similarly, other studies implied that exogenously administered L-arginine decreased oxidative stress and had a beneficial effect on tubulointerstitial fibrosis caused by ureter obstruction [62]. Previous studies demonstrated that feeding a diet with a high dose of Larginine in rabbits was beneficial for severe PBOO [63]. On the other hand, pretreatment with L-NAME, a NOS inhibitor, significantly inhibited the generation of nitrotyrosine [43]. L-NAME treatment may reduce apoptosis induced by I/R in the bladder and significantly enhance contractile responses compared to the I/R group without L-NAME. [44,46,64]. Therefore, NO was thought to have a dual role, serving as a regulator under physiological and pathophysiological conditions. In this study, the MetS group and the MetS + OVX group showed significant deterioration in bladder capacity and enhanced bladder overactivity, whereas L-arginine (NO precursor) significantly alleviated storage dysfunction and ameliorated the symptoms of MetS with or without OHD-induced detrusor overactivity. Additionally, the expressions of NOS (iNOS, eNOS, and nNOS), HIF-1α, and NRF2 were significantly suppressed in the MetS group and the MetS + OVX group in comparison with the control group, whereas the expressions of NFκB-P65, DNP, and nitrotyrosine were significant overexpressed (Figure 8). After L-arginine treatment, the levels of NFκBP65, DNP, and nitrotyrosine were meaningfully reduced in the MetS + L-arginine group and the MetS + OVX + L-arginine group, as compared to the MetS group and the MetS + OVX group. Whereas, the levels of all NOS and transcription factors (HIF-1α and NRF2) were markedly increased. Moreover, MetS with or without OHD induced the expressions of oxidative markers (DNP and nitrotyrosine) in the bladder, whereas L-arginine alleviated the extent of these expressions. These findings demonstrated that MetS, with or without OHD, reduced bladder NO production and worsened oxidative damage through the NF-κB signaling pathway. In contrast, L-arginine provided a protective effect by reducing MetS and OHD-related oxidative damage via the NRF2/HIF-1α signaling pathway.

Experimental and clinical observations indicated MetS, IR, hyperandrogenism, chronic inflammation, and excessive oxidative stress are the main factors to cause bladder disease. Improving LUT perfusion and addressing oxidative stress could offer a new therapeutic approach for treating bladder dysfunction caused by chronic ischemia [65]. In obese mice, treatment with resveratrol or the guanylyl cyclase activator BAY 60-2770 can improve OAB by leveraging their antioxidant properties. Obese mice exhibited bladder dysfunctions, including increased voiding frequency and non-voiding contractions [66,67]. Obese mice, in comparison to lean mice, exhibit significantly higher expression of oxidative stress markers (gp91phox and SOD1), increased ROS/RNS levels in bladder tissues, and elevated serum lipid peroxidation [68]. Increased levels of C-reactive protein and specific proinflammatory cytokines have been found in the serum or urine of MetS patients, suggesting that systemic inflammation and oxidative stress contribute to the pathophysiology of OAB [69,70]. The β1-adrenoceptor antagonist, phosphodiesterase type 5 inhibitor, free radical scavengers, and β3-adrenoceptor agonist have been investigated in animal models of chronic bladder ischemia. These drugs, which enhance blood flow and reduce oxidative stress, not only preserve urodynamic parameters but also protect muscle contractility and prevent changes in the bladder wall [65,71]. Increased vascular pressure may lead to elevated eNOS expression with the involvement of NFκB [72,73,74]. Endothelial dysfunction was involved in abnormalities of vascular signaling, increased oxidative stress, and enhanced inflammatory cell infiltration, which resulted in atherosclerosis and thrombosis. In this study, the level of LDH and ALK-P was elevated in MetS with or without OHD groups, as compared to the control group. Whereas, there were significant decreases in the ratio of Alk-P to LDH (Table 1). However, treatment with L-arginine reduced those levels in the MetS + L-arginine group and the MetS + OVX + L-arginine group. Whereas, the levels of all NOS and transcription factors (HIF-1α and NRF2) were markedly increased.

The nNOS plays a crucial role in modulating synaptic plasticity, regulating blood pressure, controlling smooth muscle tone through NO production at nitrergic nerve endings, and managing blood supply to skeletal muscle fibers [75,76,77,78,79]. Alterations in urothelial receptor function, neurotransmitter release, and the sensitivity and coupling of suburothelial interstitial cells can contribute to increased involuntary bladder contractions [80]. Additionally, prolonged fructose-induced MetS results in alterations in bladder purinergic and muscarinic signaling pathways. In this study, we found MetS with or without OHD resulted in neuronal degeneration while L-arginine could enhance neurogenesis and receptor response to improve bladder overactivity in a rat model of MetS with or without OHD-induced detrusor overactivity. In addition, PDE5 inhibitors are recognized as an effective treatment for OAB. While the precise mechanism of action of PDE5 inhibitors is not fully understood, multiple studies have demonstrated that these drugs can prolong the effect of endothelial-cell-released NO, enhance blood perfusion, and oxygenation in the lower urinary tract and simultaneously relax the smooth muscles of the detrusor, prostate, and urethra [81].

This investigation had several limitations. Firstly, the combination of MetS with OHD appeared to be more complex than MetS alone, leading to less pronounced effects of L-arginine. Additionally, L-arginine treatment in the MetS + L-arginine and MetS + OVX + L-arginine groups was limited in its ability to fully restore conditions to the control level. Additionally, the assessment was limited to certain mitochondrial respiratory enzyme complexes, including SDHA, UQCRC1, COX-2, and ATPB. While these markers provide valuable insight, they do not encompass the full spectrum of mitochondrial dysfunction indicators. Future studies should consider a broader array of mitochondrial markers to fully understand the impact of MetS and OHD on mitochondrial function and bladder damage. Furthermore, the rat models used in this study, including those with MetS, OVX-induced OHD, and L-arginine treatment, provide valuable insights but may not fully replicate human physiology and pathology. Differences between rat and human models could limit the applicability of the results to clinical settings. The durability and persistence of the therapeutic effects also require long-term follow-up studies for evaluation. However, organ aging and degeneration after 12 months on a high-fat, high-sugar diet with or without OVX pose challenges to physiological assessments and protein expression measurements.

## 4. Materials and Methods

### 4.1. Animals and Feeding Protocol

The experimental procedures are approved by the Committee for the Use of Experimental Animals of Kaohsiung Medical University and adhere to the guidelines of the National Institute of Health for the use of the experimental animals. Eight-week-old female Sprague-Dawley rats were divided into seven groups: (1) a control group fed a normal rat chow diet combined with 0.9% normal saline/500–700 μL intraperitoneal; (2) a Metabolic Syndrome (MetS) group induced by a high fat high sugar (HFHS) diet combined with IP injection of 500–700 μL of 0.9% normal saline; (3) a MetS + L-arginine group, where MetS was induced by the HFHS diet combined with 2% L-arginine treatment (NO precursor, IP 2 mg/kg); (4) a MetS + L-NAME group, where MetS was induced by the HFHS diet combined with L-NAME treatment (NO inhibitor, IP 10 mg/kg); (5) a MetS + OVX group, where MetS was induced by the HFHS diet combined with bilateral ovariectomy (OVX) and IP injection of 500–700 μL of 0.9% normal saline; (6) a MetS + OVX + L-arginine group, where MetS was induced by the HFHS diet combined with bilateral OVX and L-arginine treatment; and (7) a MetS + OVX + L-NAME group, where MetS was induced by the HFHS diet combined with bilateral OVX and L-NAME treatment. Rats were fed with a HFHS diet (DyEts Inc., Louis, MO, USA; DyEt, #D12120101) to induce MetS and received OVX surgery to deprive ovarian hormones. By dieting with a HFHS for 12 months, rats developed MetS and OAB. Both ovaries were excised through bilateral abdominal incisions and then closed with 2-0 silk. Bilateral OVX was performed under halothane anesthesia and every effort was made to minimize suffering and the number of animals used in the experiment. In the text, OVX stands for bilateral ovariectomy. OVX is described in further detail below when it is mentioned. When intraperitoneal injecting drugs into rats, 0.9% normal saline/500–700 μL was used to dilute L-arginine (IP 2 mg/kg) and L-NAME (IP 10 mg/kg). As the rats gained weight, the dosage of 0.9% normal saline, L-arginine, and L-NAME was adjusted. For example, the control group weighing 301–400 g and 401–500 g was fed a normal rat chow diet combined with intraperitoneal 0.9% normal saline/500 μL and 600 μL. Moreover, the MetS + L-arginine group weighing 400 g and 500 g was fed a HFHS diet combined with intraperitoneal 0.9% normal saline/500 μL and 0.8 mg L-arginine as well as 600 μL and 1.0 mg L-arginine. The MetS + L-NAME group weighing 400 g and 500 g was fed a HFHS diet combined with intraperitoneal 0.9% normal saline/500 μL and 4.0 mg L-NAME as well as 600 μL and 5.0 mg L-NAME. However, as the rats gained weight, the dosage of normal saline was adjusted. Cystometrogram study and tracing analysis of voiding behavior were used to identify the symptoms of detrusor hyperactivity. The experimental procedure is depicted in Figure 11.

### 4.2. Ovariectomy Procedure

After a 1-week acclimation period, anesthesia was induced in the rats using isoflurane (2–3%). The dorsal area was then shaved and sterilized with 75% ethanol. To expose the ovaries, two dorsolateral incisions were made, followed by ligation and the removal of both ovaries. The muscle and skin layers were then sutured in layers and disinfected. Postoperatively, if discomfort occurred, analgesic Ketoprofen (0.2–0.5 mg/100 g) was administered via subcutaneous injection.

### 4.3. Estradiol Concentration by Enzyme-Linked Immunosorbent Assay (ELISA)

Four weeks after OVX surgery, blood for estradiol measurements was collected from the tail vein under anesthesia. At the end of the experiment, 1 mL of blood was collected and centrifuged at 4 °C for serum estradiol analyses. The 17-β estradiol ELISA kit (Cayman Chemical Co., Ann Arbor, MI, USA) utilized microtiter wells coated with an antibody specific to an estradiol molecule. After the substrate solution was added, the color intensity developed as inversely correlated with the estradiol concentration in the rat samples, as measured by the ELISA (Bio-Tek ELX 800, BioTek, Bad Friedrichshall, Germany). Mean absorbance values were calculated for each set of standards and serum samples from the experimental rats. A standard curve was generated by plotting mean absorbance against standard concentrations, which allowed for the direct determination of sample concentrations based on the absorbance values [32].

### 4.4. Physical Indicator

A non-invasive blood pressure measurement for systolic blood pressure (BP) in rats was performed by measuring the caudal ventral artery using a pulse transducer with a physiograph and an appropriate rat tail cuff. BP was subsequently measured using a non-invasive blood pressure (NIBP) machine from ADInstruments (IN125NIBP controller, New South Wales, Australia). Twelve months after feeding, systolic, diastolic, and mean arterial pressure (MAP) were measured following a 15 to 30 min equilibrium period and blood pressure was recorded for 5 min. Additionally, MAP was calculated as 1/3 systolic pressure +2/3 diastolic pressure. Physical indicators, including body weight, bladder weight, and waist circumference, were also recorded.

### 4.5. Evaluation of Liver and Renal Biochemistry Parameters

To characterize the changes in physical indicators and analysis of liver and renal biochemistry parameters in each group, blood samples were collected for biochemical analysis. Serum activity GOT and GPT and concentrations of triglycerides, cholesterol, LDL, HDL, glucose, and insulin were determined by using an automated analyzer (Selectra Junior Spinlab 100, Vital Scientific, Dieren, The Netherlands; Spinreact, Girona, Spain) according to the manufacturers’ instructions. GOT and GPT contents were analyzed for the evaluation of liver function. Standard controls were run before each determination and the values obtained for the different biochemical parameters were always within the expected ranges. The contents of serum creatinine, LDH, and ALK-P were also measured.

Urine creatinine was measured by colorimetric assay using the MeDiPRO creatinine kinase test and was calculated for evaluation of renal function [82]. According to the National Kidney Foundation, we used spot urine samples to assess proteinuria because they are convenient and reliable, serving as an accurate indicator of 24-h proteinuria [83]. Urine total protein was quantified using a colorimetric assay with Pyrogallol red dye (Wako Diagnostics and Chemicals USA Inc, Richmond, VA, USA). The Creatinine Clearance Rate (CCR) is determined from urine creatinine concentration, serum creatinine concentration, and total urine volume over 24 h. The CCR reflects the volume of blood filtered by the kidneys per minute and is used to evaluate kidney function. The calculation formula is CCr = (urine creatinine × 24 h total urine volume)/(serum creatinine × 1440) (mL/min).

### 4.6. Measure the Micturition Volume and Frequency by Physiological Metabolic Cage

After 12 months of treatment, the rats were housed individually in metabolic cages (R-2100; Lab Products, Rockville, MD, USA) and given a 24-h acclimation period under consistent conditions. Once acclimated, measurements of water intake volume were taken and urine output was collected. Micturition frequency over 24 h was monitored using a cup designed for the transducer (MLT 0380, ADI Instruments, Colorado Springs, CO, USA). Water intake and urine output volumes were also recorded and analyzed over a 3-day period.

### 4.7. Cystometrograms Studies

Cystometrograms (CMGs) were performed as previously described [84]. Briefly, rats were anesthetized with Zoletil50 (1 mg/kg i.p.). The bladder catheter (PE50 tube) was connected to a syringe pump (KD Scientific 100, KD Scientific, Holliston, MA, USA) and a pressure transducer (MLT 0380, ADI Instruments, Colorado Springs, CO, USA). This catheter was used for both bladder filling and pressure measurement. Prior to each cystometrogram (CMG), the bladder was emptied, and saline was infused at a constant rate of 0.08 mL/min. The pressure was monitored using a small-volume pressure transducer positioned with the catheter. Voiding contractions were detected by pressure increases resulting in urine leakage. CMG was recorded until bladder pressure stabilized, with at least five filling/voiding cycles per rat. Pressure and force signals were amplified (ML866 PowerLab, ADI Instruments), recorded, and digitized at 1000 samples/s using LabChart 7 software. The CMG variables recorded included the filling pressure, micturition pressure, micturition interval, voiding volume, and the presence or absence of non-voiding contractions.

### 4.8. Bladder Muscle Strips for Bladder Contractility Studies

The muscle strip contractility was measured by the stimulation of EFS, carbachol, ATP, and KCl. Bladder longitudinal strips (about 5 × 15 mm^2^) were obtained from the bladder trigone to the dome. The strips were placed in oxygenated Krebs-Henseleit solution under the temperature of 37 °C for 30 min. An initial resting tension of 2 g was applied for 30 min. The strips were stimulated by an electrical field at 2, 8, and 32 Hz, followed by carbachol (20 μM), ATP (4 mM), and KCl (120 mM). The data were digitized and analyzed by the Grass POLYVIEW A-D and conversion system (Grass Instrument Co, Warwick, RI, USA).

### 4.9. Masson’s Trichrome Staining for Morphological Change

After cystometric studies, experimental rats were perfused with saline through the left ventricle to clear blood from the tissues. The bladders were then excised using forceps and sterile blades. Bladders were sectioned sagittally for bladder contractility studies and horizontally for morphological analysis. The tissues were fixed in 4% paraformaldehyde for 24 h at 4 °C, embedded in paraffin, and sectioned into 5 μm slices using a microtome. The deparaffinized sections were stained with Masson’s trichrome stain (Masson’s Trichrome Stain Kit, DAKO, Glostrup, Denmark), which stains connective tissue blue and smooth muscle (SM) red. Cross-sections were photographed at 100× magnification in 10 random fields to encompass the full thickness of the bladder wall. Using Image-Pro Plus 6.0 software (Media Cybernetics Inc., Rockville, MD, USA), the areas of collagen and SM were quantified, and the ratio of collagen to SM was calculated [80].

### 4.10. Western Blot Analysis for Protein Expression

For protein isolation and Western Blot analysis, frozen tissue samples of the bladder were homogenized on ice in the buffer (50 mom Tries, pH 7.5, 5% Tiron-X100) containing the Halt Protease Inhibitor Cocktail (Pierce, Rockford, IL, USA) at 100 mg/mol and centrifuged with 14,000× *g* at 4 °C for 20 min. Protein concentration was measured using a bicinchoninic acid (BCA) assay (Pierce, Rockford, IL, USA) with a BSA standard solution and a SpectraMAX Plus microplate reader (Molecular Devices, Sunnyvale, CA, USA). Equal amounts of total bladder protein (20 μg) were loaded onto 12% SDS-PAGE gels and transferred to PVDF membranes (Immobilon-P, Millipore, MA, USA) using Towbin buffer. After being blocked with 5% non-fat milk, the membrane was incubated with the primary antibody for inflammatory and fibrotic markers (TGF-β1, fibronectin, and collagen I), urothelial markers (E-Cadherin and CK14), interstitial markers (C-Kit, PDGFR, and vimentin), neurogenesis-related markers (neurofilament, NeuN, GFAP, muscarinic receptors (M2 and M3), and purinergic receptors (P2X3 and P2X7)), angiogenesis markers (α-SMA, Laminin, and VEGF), nitric oxide synthase (NOS) (iNOS, eNOS, and nNOS), oxidative stress markers (nitrotyrosine, 2,4-dinitrophenol (DNP)), mitochondrial respiratory enzyme complexes (NDUFS3, SDHA, UQCRC2, COX-2, and ATPB), and transcription factors (HIF-1α, NRF2, and NFκB-P65). The obtained results were normalized with glyceraldehyde-3-phosphate dehydrogenase (GAPDH) and β-Actin. Other materials and procedures used in Western Blot experiments are described in Appendix A.

Blots were visualized with enhanced chemiluminescence (ECL) and exposed to Biomax L film (Kodak, Rochester, NY, USA). Negative controls and molecular weight markers were used to identify non-specific bands. After primary antibody incubation, membranes were washed with TTBS and incubated with the goat anti-mouse IgG secondary antibody. Unbound antibodies were removed with additional TTBS washes. Protein bands were detected using ECL-Plus (Amersham Pharmacia Biotech, Freiburg, Germany) and analyzed with a Kodak Image Station 440CF and ImageJ 1.48j software. All Western Blots were performed in triplicate.

### 4.11. Immunofluorescence Staining to Detect the Location of Protein Expression

Bladder tissues were fixed overnight in 4% paraformaldehyde in 0.1 M, pH 7.4 PBS, embedded in paraffin, and sectioned into 5 µm slices. Double immunofluorescence staining was then used to localize target proteins. [32,33]. Bladder sections were blocked with 10% NGS in PBS/0.5% Triton X-100 for 1 h, then incubated overnight at 4 °C with primary antibodies against E-Cadherin (Abcam, Cambridge, MA, USA, rabbit polyclonal IgG, 1:100), Neurofilament (Novus, Littleton, CO, USA, mouse monoclonal IgG, 1:100), and α-SMA (Abcam, rabbit polyclonal IgG, 1:100). After washing, sections were treated with secondary antibodies (Invitrogen, Carlsbad, CA, USA, 1:800) at room temperature for 1 h, followed by DAPI staining and mounting with Prolong Gold anti-fade reagent. Negative controls without primary antibodies were included to identify non-specific immunostaining.

### 4.12. Statistical Analysis

Statistical analysis included ANOVA followed by the Bonferroni post hoc test and two-way ANOVA for individual comparison. The mean, standard deviation (SD), and *p* values were calculated from three independent experiments. The analysis aimed to: (1) compare treatment and control groups to assess the effect of NO on detrusor overactivity in MetS and OHD rat models, with significance indicated by * *p* < 0.05 and ** *p* < 0.01; (2) investigate potential mechanisms of NO-mediated oxidative stress by comparing the MetS + L-arginine group with the MetS group, with significance marked by ^†^ *p* < 0.05 and ^††^ *p* < 0.01; (3) evaluate the effect of NO on bladder repair post-OVX under MetS conditions by comparing the MetS + OVX group with the MetS + OVX + L-arginine group, with significance denoted by ^#^ *p* < 0.05 and ^##^ *p* < 0.01.

## 5. Conclusions

In summary, both MetS and OHD, independently or in combination, resulted in decreased nitric oxide (NO) production and reduced angiogenesis. These changes led to an increase in oxidative stress, contributing to bladder overactivity through activation of the NF-κB signaling pathway. In contrast, L-arginine treatment improved detrusor overactivity symptoms and mitigated oxidative damage by enhancing the NRF2/HIF-1α signaling pathway. This was demonstrated in a rat model of MetS with or without OHD-induced overactive bladder (OAB), highlighting L-arginine’s potential therapeutic benefits in counteracting the adverse effects of MetS and OHD on bladder function.

## Figures and Tables

**Figure 1 ijms-25-11103-f001:**
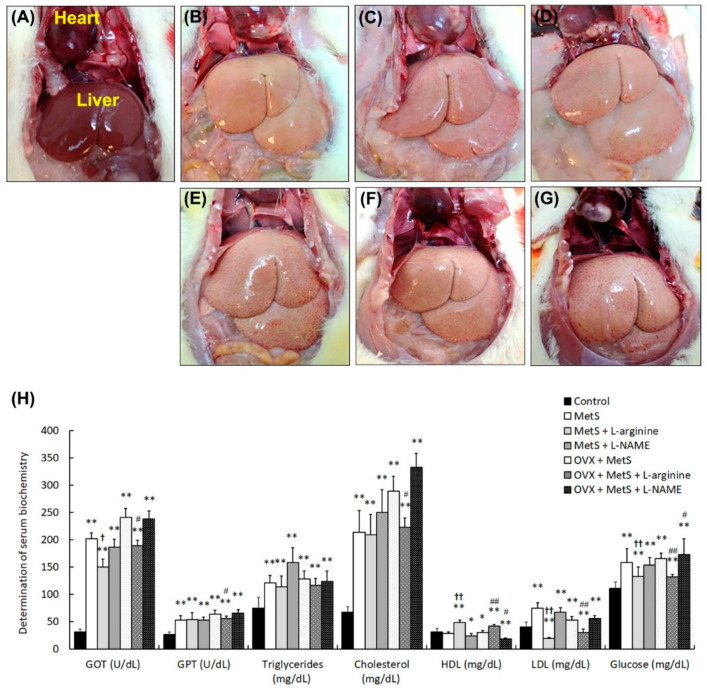
L-arginine treatment ameliorated fatty liver and improved serum parameters in rats. The changes in liver morphology (**A**–**G**) and physical indicators (**H**) after 12 months of standard diet feeding (control group, **A**) and HFHS diet feeding without OVX in the MetS group (**B**), the MetS + L-arginine group (**C**), and the MetS + L-NAME group (**D**) were shown, as were HFHS diet feeding with OVX in the MetS + OVX group (**E**), the MetS + OVX + L-arginine group (**F**), and the MetS + OVX + L-NAME group (**G**). (**A**–**G**): The control group (**A**) exhibited a dark red liver appearance. However, the liver photographs after MetS with or without OVX displayed a fatty and swollen liver appearance. Particularly, liver appearance in the MetS + OVX group (**E**) and the MetS + OVX + L-NAME group (**G**) was obviously fatty and edematous. However, the appearance of the MetS + L-arginine group showed signs of improvement compared to other groups, suggesting a beneficial effect of L-arginine treatment. (**H**): Serum parameters were significantly elevated in the MetS with or without HFHS diet feeding groups as compared to the control group, including GOT, GPT, triglycerides, cholesterol, LDL, glucose, and LDH (except insulin level). Treatment with L-arginine decreased the levels in the MetS + L-arginine group and the MetS + OVX + L-arginine group. Note: GOT, glutamate oxaloacetate transaminase; GPT, glutamate pyruvate transaminase; HDL, high-density lipoprotein; LDL, low-density lipoprotein; OHD, ovarian hormone deficiency. Data were expressed as mean ± SD for *n* = 6. * *p* < 0.05; ** *p* < 0.01 versus the control group. ^†^
*p* < 0.05; ^††^
*p* < 0.01 versus the MetS group. ^#^
*p* < 0.05; ^##^
*p* < 0.01 versus the MetS + OVX group.

**Figure 2 ijms-25-11103-f002:**
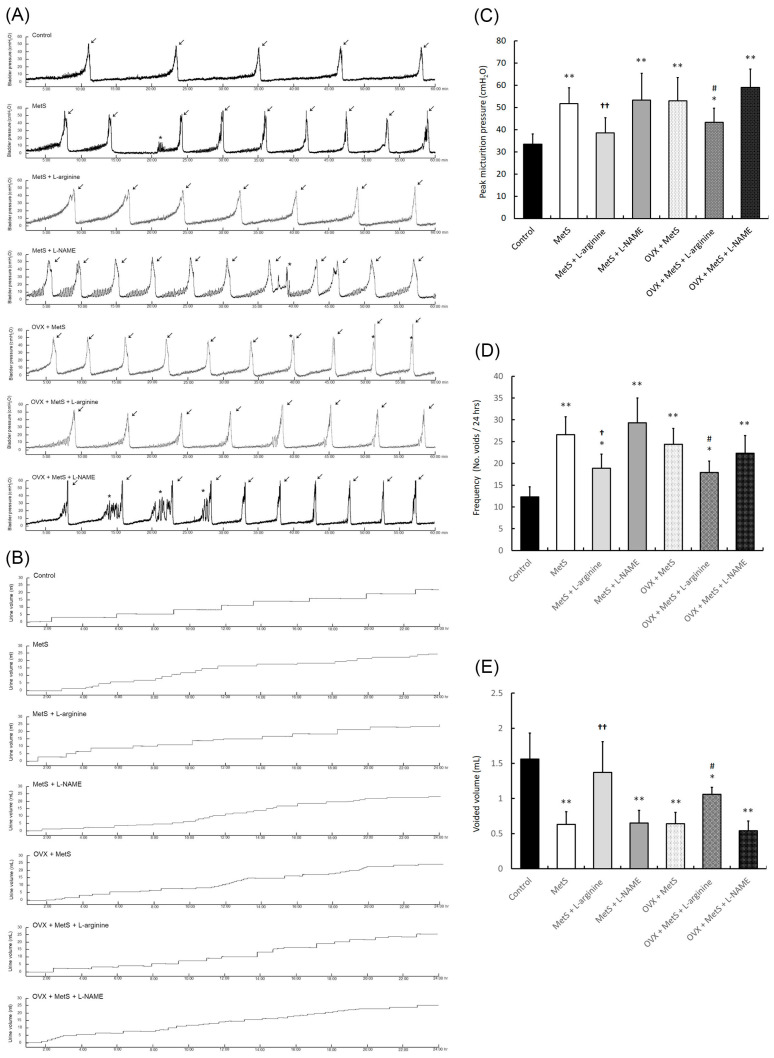
L-arginine improved voiding behavior and alleviated detrusor hyperactivity in a rat model. Urodynamic analysis of cystometric parameters (**A**), including micturition pressure (**A**,**C**), voiding frequency, contraction (arrows), and non-voiding contraction (asterisks), in the different groups. Tracing analysis of 24-h voiding behavior by metabolic cage, including voiding frequency (**B**,**D**) and volume (**B**,**E**) in the different groups. The MetS + OVX group exhibited increased bladder micturition pressure, voiding contractions, non-voiding contractions, and micturition frequency, whereas the L-arginine groups showed an improved bladder voiding pattern and volume. Note: MetS, metabolic syndrome; OHD, ovarian hormone deficiency. Data were expressed as mean ± SD for *n* = 6. * *p* < 0.05; ** *p* < 0.01 versus the control group. ^†^ *p* < 0.05; ^††^ *p* < 0.01 versus the MetS group. ^#^ *p* < 0.05 versus the MetS + OVX group.

**Figure 3 ijms-25-11103-f003:**
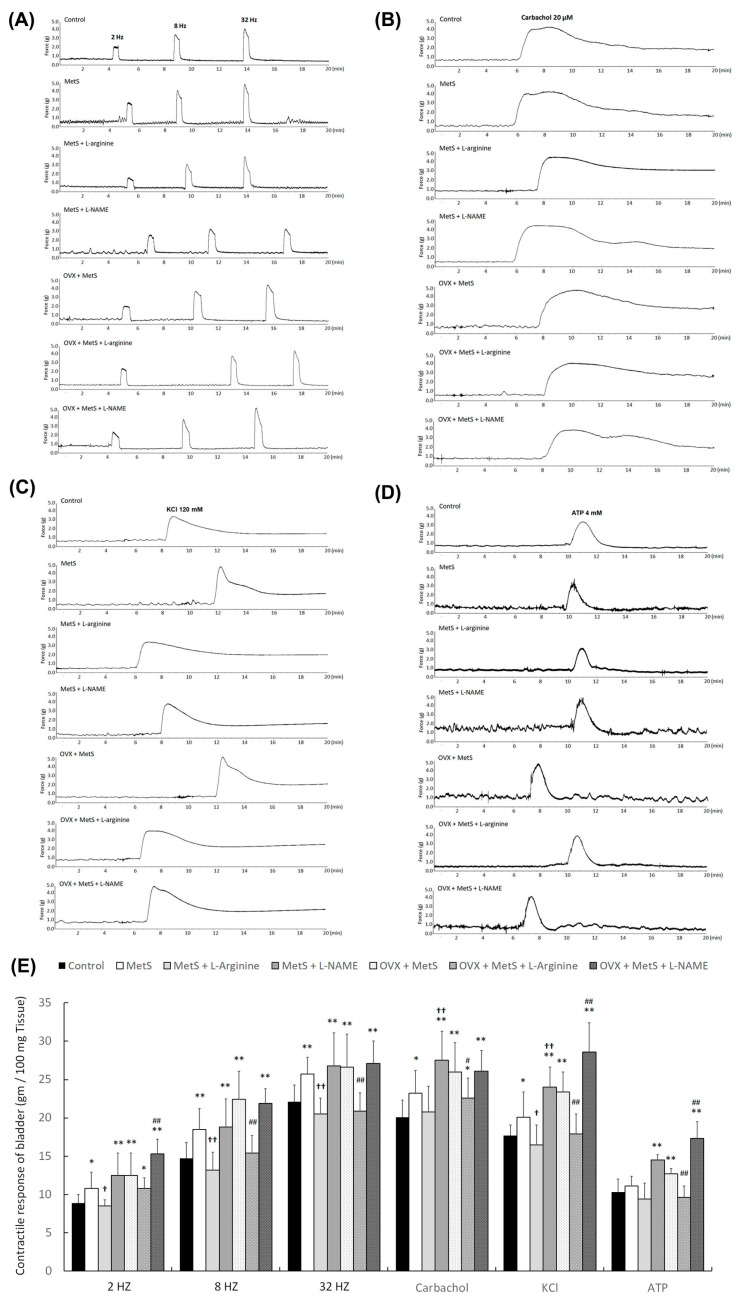
L-arginine treatment improved the bladder detrusor contractile response. After 12-month HFHS feeding with or without OVX, bladder strips induced by EFS (**A**,**E**), carbachol (**B**,**E**), KCl (**C**,**E**), and ATP (**D**,**E**) in the MetS group, the MetS + L-NAME group, the MetS + OVX group, and the MetS + OVX + L-NAME group, they had higher contractile responses compared with the control group, whereas the MetS + L-arginine and MetS + OVX + L-arginine groups demonstrated significantly lower contractile responses compared to the MetS and MetS + OVX groups. L-arginine treatment significantly ameliorated the detrusor contractile response to various forms of stimulation in the MetS + L-arginine group and the MetS + OVX + L-arginine group. Note: EFS, electrical field stimulation; OVX, bilateral ovariectomy. Data were expressed as mean ± SD for *n* = 6. * *p* < 0.05; ** *p* < 0.01 versus the control group. ^†^ *p* < 0.05; ^††^ *p* < 0.01 versus the MetS group. ^#^
*p* < 0.05; ^##^ *p* < 0.01 versus the MetS + OVX group.

**Figure 4 ijms-25-11103-f004:**
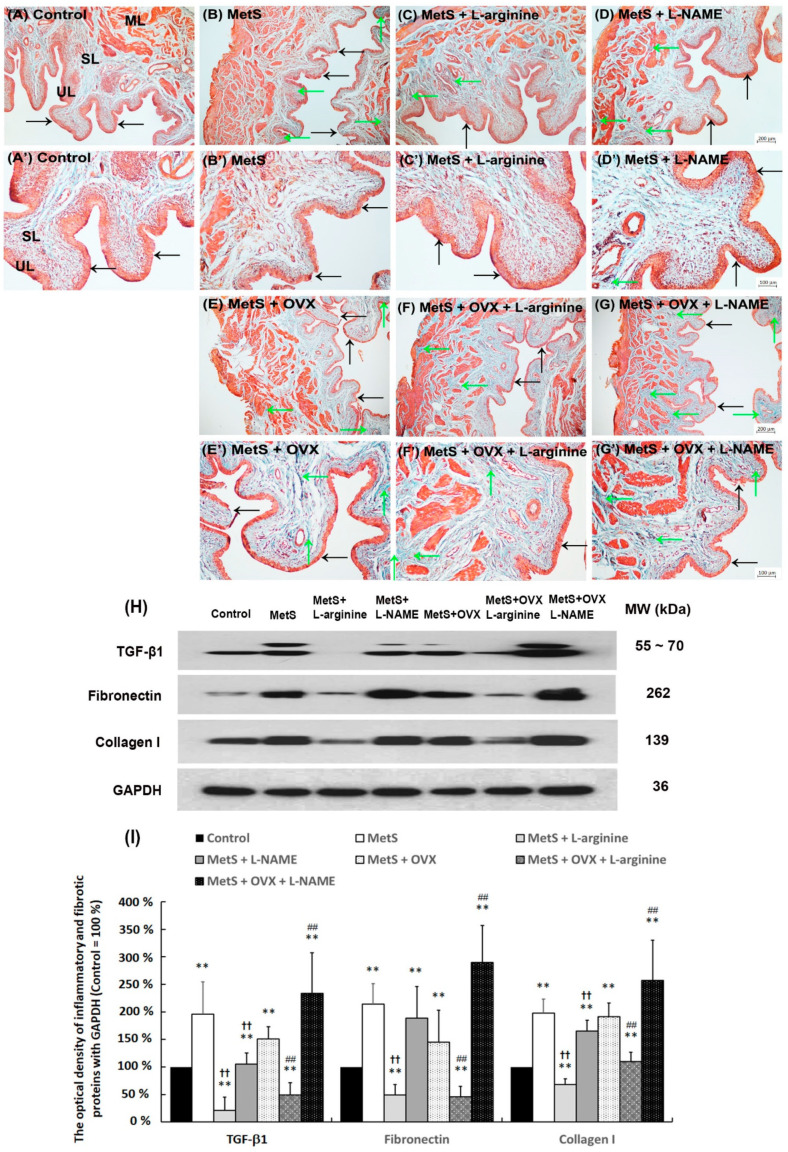
The bladder pathological features induced by the HFHS diet and OHD were shown by Masson’s trichrome staining and fibrosis marker expressions. (**A**–**G**): Bladder pathological features of the control group (**A**,**A’**), the MetS group (**B**,**B’**), the MetS + L-arginine group (**C**,**C’**) the MetS + L-NAME group (**D**,**D’**), the MetS + OVX group (**E**,**E’**), the MetS + OVX + L-arginine group (**F**,**F’**), and the MetS + OVX + L-NAME group (**G**,**G’**). Masson’s trichrome staining revealed red-stained smooth muscle and green-stained collagen. In the control group (**A**), there were three to five layers of the urothelium (UL, black arrows) with sparse collagen (green arrows) distributed in the submucosal layer (SL, lamina propria). In the MetS + OVX group (**B**), the morphology was characterized by a thinner UL (black arrows) and increased interstitial fibrosis (green arrows). In contrast, the MetS + L-arginine group (**C**) and the MetS + OVX + L-arginine group (**F**) exhibited an improved bladder condition with a thicker UL (black arrows) and reduced interstitial fibrosis (green arrows), compared to the MetS group (**B**) and the MetS + OVX group (**E**). The MetS + OVX group also showed increased bladder fibrosis (arrows), denuded urothelial mucosa (arrowheads), and a thinning UL. Therapeutic effects of NO improved pathological alteration induced by MetS with or without OHD. (**A**–**G**), magnification × 200; Scale bar (grey) = 200 μm.; (**A’**–**G’**), magnification × 400; Scale bar (grey) = 100 μm. (**H**,**I**): Western Blots for fibrosis marker expression were measured by TGF-β1, fibronectin, and type I collagen in each group. Compared to the control group, the expression of TGF-β1, fibronectin, and type I collagen proteins was significantly elevated in the MetS group compared to the MetS + L-arginine group, as well as in the MetS + OVX group compared to the MetS + OVX + L-arginine group. Therefore, L-arginine administration greatly decreased fibrosis marker expression. Quantifications of the percentage of TGF-β1, fibronectin, and type I collagen expressions to β-actin were shown. Results were normalized as the control = 100%. Note: MetS, metabolic syndrome; ML, muscular layer; OVX, bilateral ovariectomy; OHD, ovarian hormone deficiency; SL, suburothelial layer; UL, urothelial layer. Data were expressed as mean ± SD for *n* = 6. ** *p* < 0.01 versus the control group. ^††^
*p* < 0.01 versus the MetS group. ^##^ *p* < 0.01 versus the MetS + OVX group.

**Figure 5 ijms-25-11103-f005:**
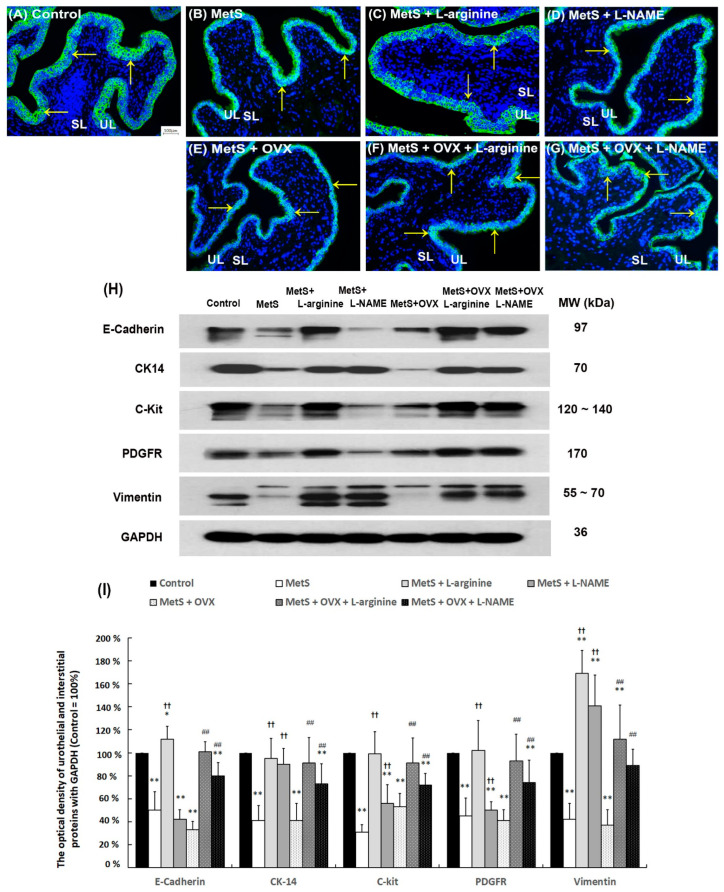
L-arginine improved bladder urothelial regeneration and interstitial cell generation. In a rat model of MetS with or without OHD-induced detrusor hyperactivity, urothelial marker (E-Cadherin), cell proliferating proteins (CK14), and IC markers (C-Kit, vimentin, and PDGFR) were quantified by immunostaining (**A**–**G**) and Western Blots (**H**,**I**). (**A**–**G**): In the control group (**A**), E-Cadherin staining showed the urothelial layer (UL; yellow arrows) consisting of three to five layers. However, following a HFHS diet with or without OVX, the bladders displayed a thinner and defective urothelial mucosa in the UL. Morphological evaluation in the MetS + L-arginine group (**C**) and the MetS + OVX + L-arginine group (**F**) showed an increased thicker layer of UL to improve bladder damage induced by MetS with or without OHD. (**A**–**G**) magnification × 400; Scale bar (grey) = 100 μm. (**H**,**I**): Western Blot analysis of E-Cadherin, CK14, C-Kit, vimentin, and PDGFR expressions was investigated. All expressions in the MetS group and the MetS + OVX group were significantly declined as compared with the control group, whereas all expressions in the MetS + L-arginine group and the MetS + OVX + L-arginine group were significantly enhanced compared to the MetS group and the MetS + OVX group. Results were normalized as the control = 100%. Note: IC, interstitial cell; MetS, metabolic syndrome; ML, muscular layer; UL, urothelial layer; SL, suburothelial layer; OVX, bilateral ovariectomy; OHD, ovarian hormone deficiency. Data were expressed as mean ± SD for *n* = 6. * *p* < 0.05; ** *p* < 0.01 versus the control group. ^††^
*p* < 0.01 versus the MetS group. ^##^ *p* < 0.01 versus the MetS + OVX group.

**Figure 6 ijms-25-11103-f006:**
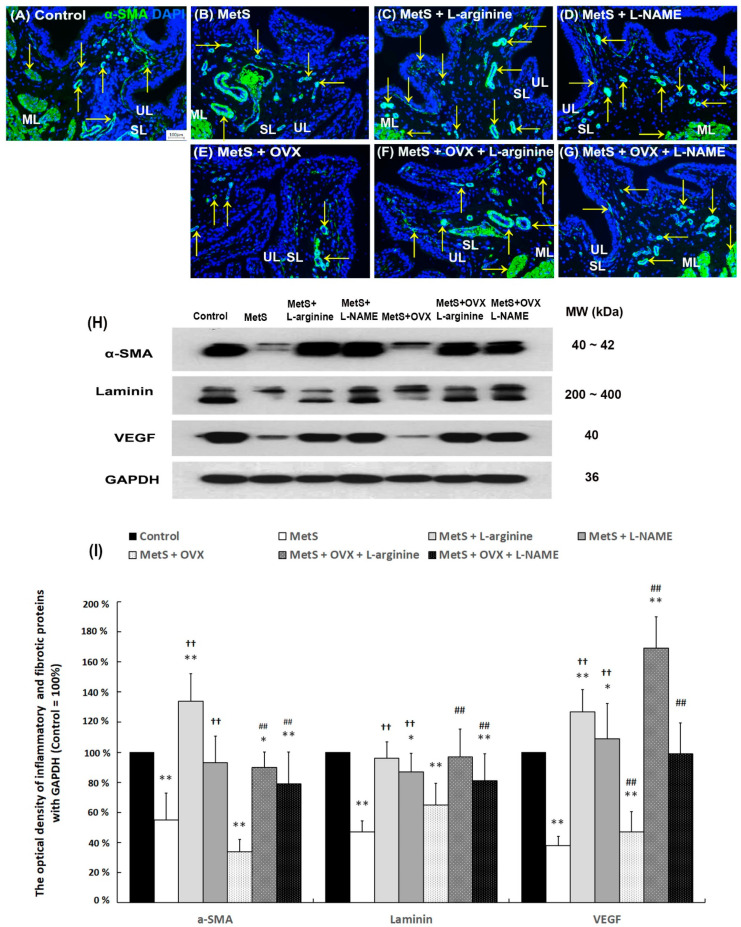
The effect of L-arginine enhanced bladder angiogenesis. (**A**–**G**): The distribution of α-SMA (yellow arrows) for angiogenesis was shown by immunostaining. In the control group (**A**), α-SMA staining (yellow arrows) was abundantly expressed on the microvasculature at SL and ML while the staining was decreased in the SL and ML of the MetS group (**B**) and the MetS + OVX group (**E**). The immunostaining of the MetS + L-arginine group (**C**) and the MetS + OVX + L-arginine group (**F**) showed an enhancement of the expression. (**A**–**G**) magnification × 400; Scale bar (grey) = 100 μm. (**H**,**I**): the protein levels of angiogenesis (α-SMA, Laminin, and VEGF) were evaluated by Western Blot analysis. The levels of α-SMA, Laminin, and VEGF markers were significantly decreased in the MetS group and the MetS + OVX group compared to the control group, whereas all expressions in the MetS + L-arginine group and the MetS + OVX + L-arginine group were significantly enhanced compared to the MetS group and the MetS + OVX group. Note: α-SMA, α-smooth muscle actin; UL, urothelial layer; SL, suburothelial layer; ML, muscular layer; OHD, ovarian hormone deficiency; PDGFR, platelet-derived growth factor receptor; VEGF, vascular endothelial growth factor. Nuclear DNA was labeled with DAPI (blue). Results were normalized as the control = 100%. Data were expressed as mean ± SD for *n* = 8, * *p* < 0.05; ** *p* < 0.01 versus the sham group. ^##^ *p* < 0.01 versus the OVX group. ^††^ *p* < 0.01 versus the OVX + SW4 group.

**Figure 7 ijms-25-11103-f007:**
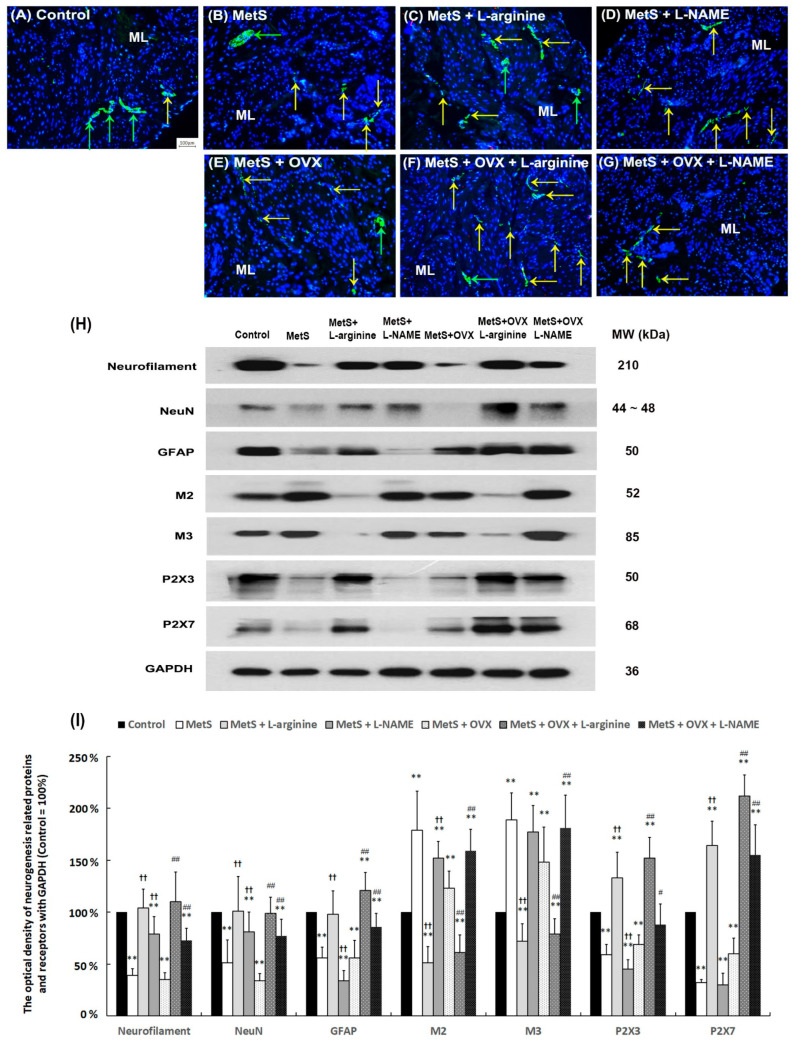
The effect of L-arginine increased neuronal regeneration, synaptic transmission, and receptor response. The expressions of neuronal endogenous markers (neurofilament, NeuN, GFAP), muscarinic receptor (M2 and M3) markers, and purinergic receptor (P2X3 and P2X7) markers were assessed by immunostaining (**A**–**H**) and Western Blots (**I**). (**A**–**G**): The distribution of neurofilament for neurogenesis was shown by immunostaining. Neurofilament immunostaining (yellow arrows) and ganglion (green arrows) were prominently expressed in the SL and ML of the control group (**A**). In contrast, the MetS group (**B**) and the MetS + OVX group (**E**) showed reduced neurofilament staining (yellow arrows) and ganglion (green arrows) in the thinner and defective urothelial mucosa of the SL and ML. However, neurofilament expression (yellow arrows) was significantly increased in the MetS + L-arginine group (**C**) and the MetS + OVX + L-arginine group (**F**) compared to the MetS group (**B**) and the MetS + OVX group (**E**). This indicates that L-arginine enhances bladder synaptic transmission, receptor response, and neurogenesis, thereby improving detrusor contractile. (**A**–**G**) magnification × 400; Scale bar (grey) = 100 μm. (**H**,**I**): Quantifications of the percentage of neurogenesis-related markers, muscarinic receptors, and purinergic receptors were evaluated by Western Blotting. Nuclear DNA was labeled with DAPI (blue). Note: NF, neurofilament; NeuN, neuronal nuclear antigen and neuron; GFAP, glial fibrillary acidic protein; ML, muscular layer. Results were normalized as the control = 100%. Data were expressed as mean ± SD for *n* = 8, ** *p* < 0.01 versus the sham group. ^#^ *p* < 0.05; ^##^ *p* < 0.01 versus the OVX group; ^††^ *p* < 0.01 versus the OVX + SW4 group.

**Figure 8 ijms-25-11103-f008:**
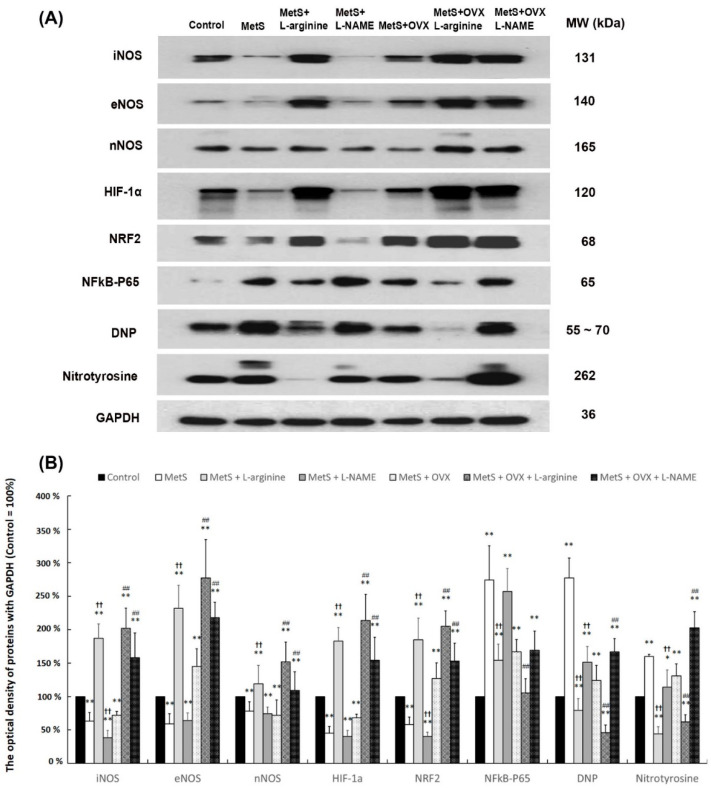
The effects of L-arginine on the expressions of oxidative stress markers in the status of MetS and OHD. (**A**) The expression levels of NOS (iNOS, eNOS, and nNOS), transcription factors (HIF-1α, NRF2, and NFkB), and oxidative stress markers (DNP and nitrotyrosine) by Western Blots. (**B**) Quantifications of the percentage of the proteins to β-actin in different experimental groups. The expression levels were significantly enhanced in the MetS group and the MetS + OVX group. Results were normalized as the control = 100%. Data were represented as mean ± SD for *n* = 6. * *p* < 0.05; ** *p* < 0.01 versus the control group. ^††^
*p* < 0.01 versus the MetS group. ^##^ *p* < 0.01 versus the MetS + OVX group.

**Figure 9 ijms-25-11103-f009:**
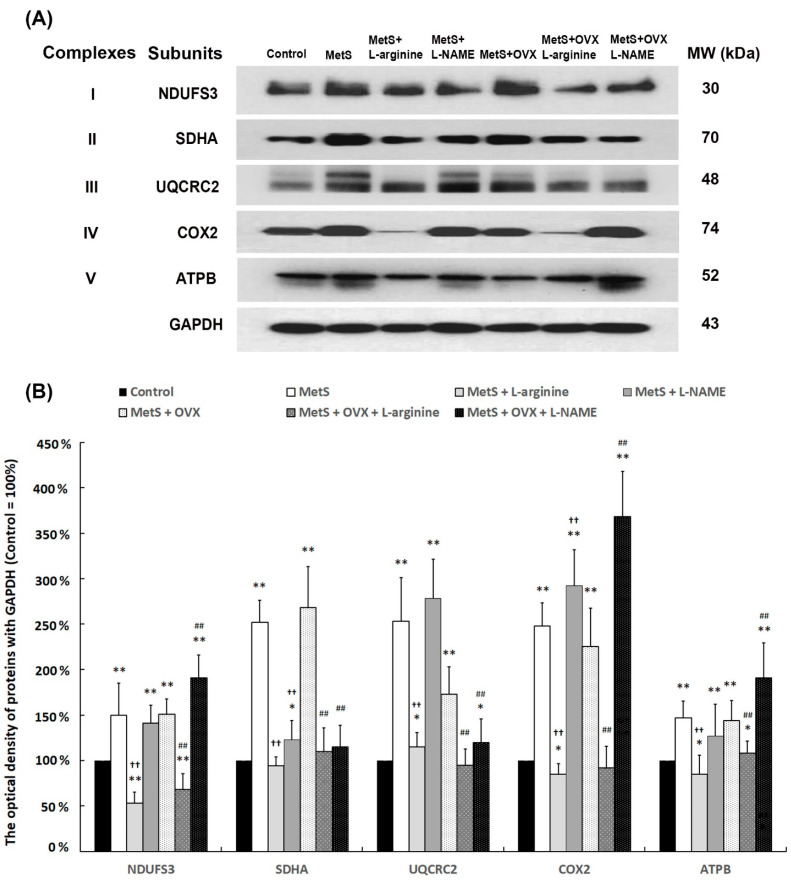
Up-regulation of the subunits of mitochondrial respiratory enzymes with MetS and OHD. (**A**) The expression levels of mitochondrial respiratory enzyme subunits (NDUFS3, SDHA, UQCRC1, COX-2, and ATPB) were analyzed by Western Blotting. (**B**) Quantification of these mitochondrial respiratory enzymes as a percentage relative to β-actin. Results were normalized to the control group, set at 100%. The expression levels of these subunits were elevated in the MetS group and significantly enhanced in both the MetS and MetS + OVX groups. Results were normalized as the control = 100%. Data were represented as mean ± SD for *n* = 6. * *p* < 0.05; ** *p* < 0.01 versus the control group. ^††^ *p* < 0.01 versus the MetS group. ^##^ *p* < 0.01 versus the MetS + OVX group.

**Figure 10 ijms-25-11103-f010:**
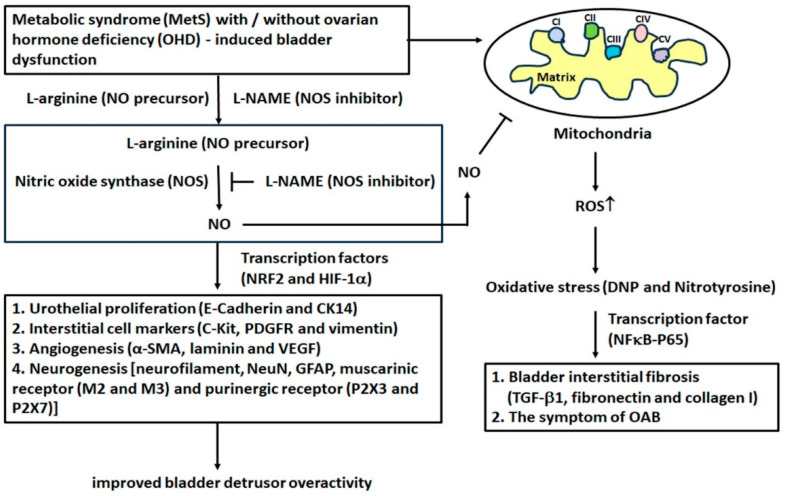
Proposed mechanistic model illustrating how MetS and OHD induce oxidative stress through mitochondria-mediated pathways and the potential mitigating effect of L-arginine on bladder overactivity. MetS and OHD induced mitochondria to release ROS to induce the generation of oxidative stress. However, L-arginine treatment reduced oxidative stress induced by MetS, with or without OHD; enhanced neurogenesis and angiogenesis; and alleviated the symptoms of OAB. Note: α-SMA, alpha smooth muscle actin; GFAP, glial fibrillary acidic protein; HIF-1α, hypoxia-inducible factor-1α; L-NAME, NG-nitro-L-arginine methyl ester; MetS, metabolic syndrome; NeuN, neuronal nuclei; NO, nitric oxide; NFκB, nuclear factor kappa-light-chain-enhancer of activated B cells; NRF2, nuclear factor erythroid 2-related factor 2; OAB, overactive bladder; OHD, ovarian hormone deficiency; OVX, bilateral ovariectomy; ROS, reactive oxygen species; TGF-β, transforming growth factor-β; TRPV, transient receptor potential vanilloid 1; VEGF, vascular endothelial growth factor.

**Figure 11 ijms-25-11103-f011:**
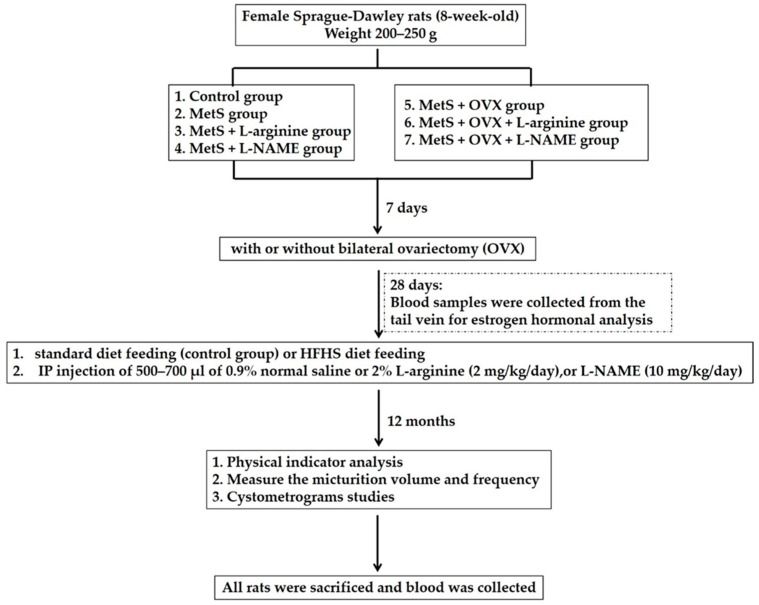
Schematic diagram of the experimental procedure.

**Table 1 ijms-25-11103-t001:** General physical indicators, urine parameters, serum biochemistry, and urodynamic parameters for the different experimental groups.

	Control	MetS	Mets + L-Arginine	MetS + L-NAME	OVX + MetS	OVX + MetS + L-Arginine	OVX + MetS + L-NAME
No. rats	15	15	12	12	15	12	12
Serum estradiol conc. (pg/mL) before OVX	40.1 ± 7.5	39.8 ± 4.5	38.9 ± 5.4	38.6 ± 6.0	38.2 ± 5.3	36.9 ± 6.2	38.8 ± 4.6
Serum estradiol conc. (pg/mL) after OVX	44.8 ± 8.5	43.5 ± 7.1	36.6 ± 5.7	38.4 ± 7.4	18.9 ± 2.1 **	19.6 ± 3.3 **	20.3 ± 3.9 **
Serum testosterone conc. (pg/mL) before OVX	60.1 ± 4.5	57.8 ± 4.5	58.9 ± 5.4	58.6 ± 6.0	58.2 ± 5.3	56.9 ± 6.2	60.8 ± 4.6
Serum testosterone conc. (pg/mL) after OVX	61.2 ± 10.9	67.2 ± 13.6	64.8 ± 12.2	61.4 ± 9.9	35.0 ± 4.7 **	36.4 ± 9.3 **	33.2 ± 4.7 **
Physical indicator							
Water intake (mL/24 h)	29.0 ± 2.6	34.2 ± 4.8	37.3 ± 2.5	28.6 ± 3.4	26.0 ± 4.2	25.0 ± 3.0	20.8 ± 3.1
Urine output (mL/24 h)	16.2 ± 1.7	17.5 ± 2.3	20.8 ± 1.9	19.0 ± 3.2	16.2 ± 1.9	17.1 ± 1.6	14.0 ± 3.9
Body weight (gm)	389.3 ± 52.7	525.9 ± 77.2 **	451.4 ± 56.7 * ^††^	500.7 ± 62.4 **	681.8 ± 65.0 **	518.1 ± 50.4 ** ^#^	692.8 ± 74.8 **
Bladder weight (mg)	155.3 ± 18.0	163.9 ± 34.3	165.8 ± 18.8 **	186.7 ± 22.2	188.2 ± 38.0	180.5 ± 30.2 **	181.1 ± 42.2 **
The ratio of bladder weight (mg)/body weight (g)	0.41 ± 0.09	0.31 ± 0.06 *	0.37 ± 0.07 ^†^	0.34 ± 0.06	0.27 ± 0.07 **	0.35 ± 0.08	0.26 ± 0.07 **
Waist circumference (cm)	18.5 ± 1.1	22.1 ± 2.8 *	21.7 ± 1.6 *	21.7 ± 1.6 *	26.4 ± 2.5 **	22.2 ± 1.3 *	24.5 ± 2.8 **
Systolic pressure (mmHg)	113.1 ± 7.9	127.2 ± 8.6 *	118.8 ± 16.0	136.2 ± 21.5 ** ^†^	131.7 ± 8.5 **	118.2 ± 10.0	139.2 ± 12.1 ** ^#^
Diastolic pressure (mmHg)	88.0 ± 5.0	97.6 ± 4.7 *	93.3 ± 15.0	118.2 ± 17.8 ** ^†^	98.7 ± 5.2 *	91.9 ± 6.4	118.2 ± 17.8 ** ^#^
MAP (mmHg)	96.4 ± 4.8	107.4 ± 5.6 *	101.8 ± 15.0	124.2 ± 19.0 ** ^††^	109.7 ± 5.9 **	100.7 ± 7.0	125.2 ± 15.5 ** ^#^
Serum biochemistry							
Creatinine (mg/dL)	0.35 ± 0.06	0.47 ± 0.09 *	0.31 ± 0.03	0.42 ± 0.05 *	0.49 ± 0.09 **	0.43 ± 0.07 *	0.48 ± 0.03 **
Insulin (Bayer) (mU/L)	0.50 ± 0.04	0.60 ± 0.08	0.54 ± 0.07	0.55 ± 0.04	0.62± 0.05	0.55 ± 0.05	0.61 ± 0.07
LDH (U/L)	157.0 ± 15.9	1433.0 ± 50.8 **	455.0 ± 60.3 ** ^††^	1834.9 ± 31.3 ** ^††^	2373.0 ± 63.3 **	585.5 ± 87.7 ** ^##^	3152.5 ± 181.2 ** ^##^
ALK-P (U/L)	63.5 ± 4.6	164.5 ± 51.5 **	102.9 ± 21.0 ** ^††^	182.7 ± 13.8 ** ^†^	280.8 ± 54.0 **	165.2 ± 31.7 ** ^##^	278.5 ± 47.9 **
The ratio of ALK-P/LDH	0.40 ± 0.10	0.11 ± 0.03 **	0.22 ± 0.06 ** ^††^	0.10 ± 0.03 **	0.11 ± 0.02 **	0.28 ± 0.05 ** ^##^	0.08 ± 0.02 ** ^#^
Urine parameters							
Glucose (mg/dL)	0	0	0	0	15.6 ± 5.2 **	9.7 ± 2.9 ** ^#^	16.6 ± 2.4 **
Urine protein (mg/dL)	16.5 ± 1.8	73.0 ± 9.3 **	38.7 ± 4.6 ** ^†^	107.3 ± 28.8 ** ^†^	74.6 ± 17.7 **	40.7 ± 11.3 ** ^##^	90.6 ± 19.2 **
The ratio of urine protein/creatinine	5.4 ± 2.0	49.2 ± 10.8 **	21.7 ± 5.3 ** ^†^	37.7 ± 5.5 **	70.2 ± 12.8 **	29.9 ± 5.7 ** ^##^	65.8 ± 13.4 **
CCR (mL/min)	1.21 ± 0.42	0.72 ± 0.11 **	1.41 ± 0.36 ^††^	0.71 ± 0.16 **	0.59 ± 0.18 **	1.58 ± 0.08 ** ^##^	0.54 ± 0.17 **
Urodynamic parameters							
Frequency (No. voids/24 h)	12.3 ± 2.3	26.6 ± 4.1 **	18.9 ± 3.2 * ^†^	29.3 ± 5.7 **	24.4 ± 3.6 **	17.9 ± 2.6 * ^#^	22.3 ± 4.1 **
Peak micturition pressure (cmH_2_O)	33.4 ± 4.6	51.8 ± 7.1 **	38.6 ± 5.9 ^††^	53.4 ± 8.2 **	53.0 ± 9.5 **	43.3 ± 6.3 * ^#^	58.9 ± 8.3 **
Voided volume (mL)	1.56 ± 0.37	0.63 ± 0.18 **	1.37 ± 0.44 ^††^	0.65 ± 0.18 **	0.64 ± 0.16 **	1.06 ± 0.10 * ^#^	0.54 ± 0.14 **
No. non-voiding contractions between micturition (No./hr)	0	0.87 ± 0.22 *	0 ^†^	0.98 ± 0.26 *	2.79 ± 0.66 **	0.85 ± 0.31 * ^#^	2.67 ± 0.43 **

ALP-P, alkaline phosphatase; CCR, creatinine clearance rate; LDH, lactic dehydrogenase; MAP, mean arterial pressure; OVX, surgical ovariectomy; Valus are mean ± SD. * *p* < 0.05; ** *p* < 0.01 versus the control group. ^†^ *p* < 0.05; ^††^ *p* < 0.01 versus the MetS group. ^#^ *p* < 0.05; ^##^ *p* < 0.01 versus the OVX + MetS group.

## Data Availability

The datasets of the present study can be available from the corresponding author upon request.

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
