# Peer review of "Effects of Nitric Oxide on Bladder Detrusor Overactivity through the NRF2 and HIF-1α Pathways: A Rat Model Induced by Metabolic Syndrome and Ovarian Hormone Deficiency"

_ijms, 2024, doi:10.3390/ijms252011103_

Round 1
Reviewer 1 Report
Comments and Suggestions for Authors
The article primarily investigates the role of nitric oxide (NO) in bladder overactivity (OAB) caused by metabolic syndrome (MetS) and ovarian hormone deficiency (OHD) using a rat model. The research reveals that MetS and OHD lead to bladder dysfunction, and these symptoms can be alleviated by the administration of L-arginine (a precursor of NO), mainly via the NRF2 and HIF-1α signaling pathways. This study provides new insights into the role of nitric oxide in the regulation of bladder function.
1. Further experimental validation needed: The role of NO in regulating the NRF2 and HIF-1α signaling pathways should be validated with knockdown or overexpression experiments.
2. Long-term follow-up: It would be beneficial to include long-term follow-up studies to assess the persistence and durability of the treatment effects.
3. Mechanism exploration: A deeper exploration of the mechanisms through which NO modulates the NRF2 and HIF-1α signaling pathways is recommended. Additionally, other potential molecular regulatory factors involved should be discussed.
Comments on the Quality of English LanguageMinor editing of English language required.
Author Response
Reviewer 1
Comments and Suggestions for Authors
The article primarily investigates the role of nitric oxide (NO) in bladder overactivity (OAB) caused by metabolic syndrome (MetS) and ovarian hormone deficiency (OHD) using a rat model. The research reveals that MetS and OHD lead to bladder dysfunction, and these symptoms can be alleviated by the administration of L-arginine (a precursor of NO), mainly via the NRF2 and HIF-1α signaling pathways. This study provides new insights into the role of nitric oxide in the regulation of bladder function.
- Further experimental validation needed: The role of NO in regulating the NRF2 and HIF-1α signaling pathways should be validated with knockdown or overexpression experiments.
Response: Thank you for your professional recommendation. While it is currently challenging to establish an animal model in vivo for direct knockdown or overexpression of NO production to study its role in regulating the NRF2 and HIF-1α signaling pathways, there are alternatives like genetically modified mice with specific knockouts of nitric oxide synthase (NOS) genes (e.g., eNOS, iNOS, or nNOS knockout mice). These models can help elucidate the specific role of NO in these pathways. However, these genetic models may not fully capture the complexity of physiological protein interactions and bladder function assessment seen in vivo.
Therefore, in our study, we used a pharmacological modulation approach involving rats fed a high-fat, high-sugar (HFHS) diet, combined with treatments using L-arginine (a precursor of NO) and L-NAME (an NOS inhibitor). This approach allows us to effectively explore the role of NO on bladder function in a physiologically relevant context.
- Long-term follow-up: It would be beneficial to include long-term follow-up studies to assess the persistence and durability of the treatment effects.
Response: We appreciate the reviewer’s suggestion regarding the need for long-term follow-up studies to assess the persistence and durability of the treatment effects. While we agree that long-term tracking is crucial, it is important to note that after 12 months on a high-fat, high-sugar (HFHS) diet with or without OVX, the aging and degeneration of organs can become significant confounding factors, interfering with physiological assessments and protein expression analysis.
To address this, we have added the following statement to the Discussion section: “The durability and persistence of the therapeutic effects also require long-term follow-up studies for evaluation. However, organ aging and degeneration after 12 months on a high-fat, high-sugar diet with or without OVX pose challenges to physiological assessments and protein expression measurements.” (please refer to page 33, lines 776-780).
- Mechanism exploration: A deeper exploration of the mechanisms through which NO modulates the NRF2 and HIF-1α signaling pathways is recommended. Additionally, other potential molecular regulatory factors involved should be discussed.
Response: Thank you for the insightful suggestion. We agree that a deeper exploration of the mechanisms by which NO modulates the NRF2 and HIF-1α signaling pathways, particularly with treatments involving L-arginine (NO precursor) and L-NAME (NOS inhibitor), is essential. To address this, we plan to utilize next-generation sequencing (NGS), proteomics, and Western blotting to identify potential pathological mechanisms in bladder tissue associated with overactive bladder. Additionally, we acknowledge the need to explore and discuss other potential molecular regulatory factors involved, which will be a focus of our future research.
Reviewer 2 Report
Comments and Suggestions for Authors
In general methods need to be better explained, statistics need to be added into text and potentially run to test for if the main factors of diet, OVX and treatment are significant and then post hocs if relevant. I do not understand whether all other groups than OVX received SHAM or not or if the groups not receiving treatment received a control injection. If not the study design is flawed. Here are my specific comments for each section.
Introduction
Line 81 – Change “Dietary” to Diets with high cholesterol or Diet high in cholesterol.
Line 87 – Change M2 to M2- and receptor instead of receptors.
Line 91-02 - In fructose-fed rats with induced detrusor overactivity,
Line 94 – M2-
Line 94-95 - Odd sentence: “…were up-regulated in those detrusor overactivity fructose-fed rats”, perhaps: were up-regulated in the fructose-fed rats with detrusor overactivity. Also compared to what?
Line 98 – In females. Also removes also, because this is the start of a paragraph and first time bringing up OHD.
Line 132 – Extra comma and space.
General comment – it may be helpful to add a figure with the processes going on and which compounds may effect metabolic syndrome and or bladder dysfunction etc. Maybe just a simplified version of your figure 10.
Results
General – could you add the statistics in the text. P-values, F-vales etc. You also need to refer to the specific figure that your statement refers to, for example (Figure 1A).
Line 229 – “MetS is more vulnerable to cause liver damage.” Odd phrasing, please rephrase.
Line 230 – “The liver photography after 12 months of HFHS feeding.” Rephrase, for example Representative images of livers of control (Figure 1A) fed a standard diet, etc. You also need to state specifically what group is liver belong to in the figure legend.
Line 232 – Refer to Figure 1H.
Line 235 – Is that period in the middle of the sentence a typo? Also state what they are elevated in relation to – the control group?
Rephrase title for figure 1. Also make it more specific, what are A-G?
Images need scale for sizing.
Methods
This section needs to be more specific, how old were the females?
Split OVX surgery description into a new section and describe in greater detail. When was OVX conducted? At what age and when in relation to the start of diet? Did all of the other groups receive a SHAM surgery? Potentially add a figure with the experimental plan.
Did the Control and Mets group get any injection to control for IP injections that the treated groups received?
At some point in the methods and results I think that you should explain the criteria for metabolic syndrome and then how you demonstrated that the rats have metabolic syndrome. Otherwise you should just call the groups HFHS.
For statistics, I feel like your main factors should be Diet, OVX and treatment and then run posthocs when relevant.
How were the rodents euthanized? Where was the blood for estradiol measurements collected? How old were the rodents at the time of euthanasia?
Methods in section 4.3 are not well described. How were they measured? What equipment was used?
Describe section 4.8 in greater detail. For example perfusion, how bladder was removed, how was it sectioned and using what equipment. Describe staining in greater detail.
Comments on the Quality of English LanguageThe English language is good, just some oddly phrased sentences that can be easily fixed.
Author Response
Reviewer 2
Comments and Suggestions for Authors
In general methods need to be better explained, statistics need to be added into text and potentially run to test for if the main factors of diet, OVX and treatment are significant and then post hocs if relevant. I do not understand whether all other groups than OVX received SHAM or not or if the groups not receiving treatment received a control injection. If not the study design is flawed. Here are my specific comments for each section.
Introduction
- Line 81 – Change “Dietary” to Diets with high cholesterol or Diet high in cholesterol.
Response: As suggested by the reviewer, we have changed “Dietary” to “Diet”. The changes made in the manuscript is marked in red font (please refer to page 2, line 81).
- Line 87 – Change M2 to M2- and receptor instead of receptors.
Response: As suggested by the reviewer, we have revised the description in the Introduction section on page 2, line 87.
- Line 91-02 - In fructose-fed rats with induced detrusor overactivity,
Response: As suggested by the reviewer, we have revised the description in the Introduction section on page 3, line 91.
- Line 94 – M2-
Response: We have revised “M2” to “M2- “ (please refer to page 3, line 93).
- Line 94-95 - Odd sentence: “…were up-regulated in those detrusor overactivity fructose-fed rats”, perhaps: were up-regulated in the fructose-fed rats with detrusor overactivity. Also compared to what?
Response: Thank you for your suggestion. We have revised as “were up-regulated in the fructose-fed rats with detrusor overactivity” (please refer to page 3, lines 94-95).
- Line 98 – In females. Also removes also, because this is the start of a paragraph and first time bringing up OHD.
Response: We have removed “In females,” and “also” on page 3, line 98.
- Line 132 – Extra comma and space.
Response: As suggested by the reviewer, we have revised “Extra comma and space”. All the changes made in the manuscript are marked in red font on page 4, line 132.
- General comment – it may be helpful to add a figure with the processes going on and which compounds may affect metabolic syndrome and or bladder dysfunction etc. Maybe just a simplified version of your figure 10.
Response: Thank you for the suggestion. Figure 10 was designed to illustrate the correlation and interaction of key impact factors as described in the manuscript. To provide a clearer understanding of our study design and experimental results, we used L-arginine (NO precursor) and L-NAME (NOS inhibitor) in an overactive bladder animal to explore the physiological mechanisms of NO's effect on bladder overactivity. Our results led us to propose a mechanistic model showing how metabolic syndrome (MetS) and ovarian hormone deficiency (OHD) induce oxidative stress through mitochondria-mediated pathways and how L-arginine potentially mitigates bladder overactivity.
Results
- General – could you add the statistics in the text. P-values, F-vales etc. You also need to refer to the specific figure that your statement refers to, for example (Figure 1A).
Response: As suggested by the reviewer, we have added the P-values in the Result section on pages 7-10.
- Line 229 – “MetS is more vulnerable to cause liver damage.” Odd phrasing, please rephrase.
Response: Thanks for your recommendation. We have revised "MetS is more vulnerable to cause liver damage" to "MetS increases the risk of liver dysfunction." in the Results section on pages 7, line 201.
- Line 230 – “The liver photography after 12 months of HFHS feeding.” Rephrase, for example Representative images of livers of control (Figure 1A) fed a standard diet, etc. You also need to state specifically what group is liver belong to in the figure legend.
Response: Thanks for your recommendation. We have revised "The liver photography after 12 months of HFHS feeding " to " The changes of liver morphology and physical indicators after 12 months of standard diet feeding (control group, Figure 1A) and HFHS diet feeding (Figures 1B-1G) were shown in Figure 1 " in the Results section on pages 7, line 202-204.
- Line 232 – Refer to Figure 1H.
Response: As suggested by the reviewer, we have revised the description in the Introduction section on page 8, line 211-212.
- Line 235 – Is that period in the middle of the sentence a typo? Also state what they are elevated in relation to – the control group?
Response: Thank you for pointing out the typo; the period in the middle of the sentence was indeed a mistake, and we have corrected it (please refer to page 8, line 214). Additionally, since Table 1 already includes the statistical values and comparisons to the control group, we have opted not to detail the extent of increase for each group within the text to keep the manuscript concise and avoid redundancy.
- Rephrase title for figure 1. Also make it more specific, what are A-G?
Response: As suggested by the reviewer, all the changes made in the Result section are marked in red font on pages 7-8, lines 202-209. We also have revised in Figure legend on page 9, lines 231-237.
- Images need scale for sizing.
Response: As suggested by the reviewer, we have added the scale bar in the Fig 4, Fig 5, Fig 6 and Fig 7.
Methods
- This section needs to be more specific, how old were the females?
Response: Thanks for your recommendation. We have revised "Female Sprague-Dawley rats" to "8-week-old female Sprague-Dawley rats " in the Materials and Methods section on page 33, line 786-787.
- Split OVX surgery description into a new section and describe in greater detail. When was OVX conducted? At what age and when in relation to the start of diet? Did all of the other groups receive a SHAM surgery? Potentially add a figure with the experimental plan.
Response: Thank you for your suggestion. We have revised the manuscript by adding a detailed description of the OVX surgery in a new section titled "4.2 Ovariectomy Procedure" (please refer to pages 35, lines 810-815). The OVX surgery was conducted on 9-week-old female Sprague-Dawley rats (weighing 200-250 g) to induce ovarian hormone deprivation. The Control group, MetS group, MetS + L-arginine group, and MetS + L-NAME group did not receive SHAM surgery. Additionally, we have included Figure 11, which illustrates the experimental procedure for clarity (please refer to page 34, lines 806-808).
- Did the Control and Mets group get any injection to control for IP injections that the treated groups received?
Response: The Control group, MetS group, and MetS + OVX group received intraperitoneal injections of 500-700 μL of 0.9% normal saline as a control for the injections received by the treated groups. The volume of normal saline was adjusted according to the rats' weight gain. We have updated the Materials and Methods section to include this information (please refer to pages 33-34, lines 788, 790, and 795, 804-805).
- At some point in the methods and results I think that you should explain the criteria for metabolic syndrome and then how you demonstrated that the rats have metabolic syndrome. Otherwise you should just call the groups HFHS.
Response: The World Health Organization (WHO) defines the essential components of metabolic syndrome (MetS) as obesity, dyslipidemia, hypertension, and glucose intolerance. Individuals with MetS have an increased risk of overactive bladder (OAB), which worsens bladder storage function. In animal models, a rat is considered to have MetS if it exhibits more than two of the following characteristics: hypertension, obesity/increased adiposity, hyperglycemia/glucose intolerance/insulin resistance, elevated triglycerides, or reduced HDL/increased total cholesterol (Kwitek 2019).
In our study, MetS with OAB was induced in Sprague-Dawley rats after 12 months on a high-fat, high-sugar (HFHS) diet with or without OVX. As shown in Table 1, physical characteristics observed after 12 months included significant increases in body weight (p < 0.05 vs. control), bladder weight, the ratio of bladder weight to body weight (p < 0.01 vs. control), waist circumference (p < 0.01 vs. control), and blood pressure (systolic pressure, diastolic pressure, and MAP) (p < 0.01 vs. control).
Furthermore, serum parameters indicative of MetS, such as GOT, GPT, triglycerides, cholesterol, LDL, HDL, glucose, creatinine, insulin, and LDH, were significantly elevated compared to the control group, as illustrated in Figure 1. Based on these physiological and biochemical indicators, we classified these rats as having metabolic syndrome.
Reference:
- Kwitek, A. E. (2019). "Rat Models of Metabolic Syndrome." Methods Mol Biol 2018: 269-285.
- For statistics, I feel like your main factors should be Diet, OVX and treatment and then run posthocs when relevant.
Response: Thank you for your professional insights. We agree that diet, OVX, and treatment are the major factors in this study. The high-fat, high-sugar (HFHS) diet was used to induce MetS and overactive bladder (OAB), and OVX-treated rats were used to mimic postmenopausal status, which induces detrusor hyperactivity. This approach aligns with findings in postmenopausal women with ovarian hormone deficiency (OHD), who often exhibit urinary dysfunctions, including OAB symptoms. We have carefully analyzed the results considering these factors to better understand the underlying pathological mechanisms and have applied post hoc analyses when relevant to identify significant differences.
- How were the rodents euthanized? Where was the blood for estradiol measurements collected? How old were the rodents at the time of euthanasia?
Response: In the Sprague-Dawley rat model, the rats' weight was measured before anesthesia was administered. Anesthesia was given either by intraperitoneal injection of a mixed anesthetic (ketamine 8.7 mg/100 g and xylazine 1.3 mg/100 g) based on body weight or using inhaled anesthetics such as Isoflurane (2-3%) or Halothane (Induction: 1-3%; Maintenance: 0.5-1.5%) via an anesthesia machine. After anesthesia, the surgical site was shaved and disinfected with iodine. If OVX rats exhibited discomfort post-surgery, analgesic Ketoprofen (0.2-0.5 mg/100 g) was administered subcutaneously. Every effort was made to minimize suffering. Animals showing signs of weight loss, loss of appetite, weakness, or organ infection after surgery were humanely euthanized using carbon dioxide.
Four weeks after OVX surgery, blood for estradiol measurements was collected from the tail vein under anesthesia. At the end of the experiment, 1 mL of blood was collected and centrifuged at 4°C for serum estradiol analysis.
The study used 8-week-old female Sprague-Dawley rats. After 12 months (50 weeks) of feeding with an HFHS diet, with or without OVX, MetS and OAB were induced. Micturition volume and frequency were measured using a physiological metabolic cage and cystometrograms (CMGs) over 4 weeks. The rats were approximately 67 weeks old at the time of euthanasia. It is noted that aging and obesity contributed to organ degeneration, which may interfere with physiological assessments and protein expression.
- Methods in section 4.3 are not well described. How were they measured? What equipment was used?
Response: As suggested by the reviewer, we have revised the description in the Methods section (please refer to page 35, lines 831-837). We added the following: “Non-invasive blood pressure measurement for systolic BP in rats was performed by measuring the caudal ventral artery using a pulse transducer with a physiograph and an appropriate rat tail-cuff. BP was subsequently measured using a non-invasive blood pressure (NIBP) machine from ADInstruments (IN125NIBP controller, Australia).”
Although intraarterial cannulation is considered the most physiological method for blood pressure (BP) recording in rats, but it is challenging and time-consuming to maintain the patency of the arterial catheter for long-term experiments.
- Describe section 4.8 in greater detail. For example perfusion, how bladder was removed, how was it sectioned and using what equipment. Describe staining in greater detail.
Response: As suggested by the reviewer, we have revised the description in the Methods section to provide greater detail. We have added the following: “After cystometric studies, experimental rats were perfused with saline through the left ventricle to clear blood from the tissues. The bladders were then excised using forceps and sterile blades. Bladders were sectioned sagittally for bladder contractility studies and horizontally for morphological analysis. The tissues were fixed in 4% paraformaldehyde for 24 hours at 4°C, embedded in paraffin, and sectioned into 5 μm slices using a microtome.” (please refer to page 37, lines 900-905).
Comments on the Quality of English Language
- The English language is good, just some oddly phrased sentences that can be easily fixed.
Response: Thank you for your feedback on the language quality. We have reviewed and rephrased the sentences as per your suggestions to improve clarity and readability throughout the manuscript.

Round 2
Reviewer 2 Report
Comments and Suggestions for Authors
While the authors have addressed certain comments from the last round they have not addressed my comments regarding the statistical results. In addition, there are some requests were not carried out fully, for example you added the letters for each photo of the liver as requested but you do not state what group each photo is a representative image of. In addition, you added scale bars in certain figures but not others and they are teeny tiny.
Introduction
Line 81 – dietS not diet
Results
Please add the descriptive statistics for each group, F-statistic, degrees of freedom, p-value, effect size. When adding p-value in text please add the exact p-value not the definition of the asterix. In addition, you stated that you were analyzing based on diet, OVX and treatment, but you are not indicating anything about what were results were? Were there any interactions between these factors? Was the main effect of OVX, diet or treatment significant? When are you providing us with the results of post hoc analysis, none of this is described.
You need to describe these statistics: Table I shows OVX significantly decreased serum estradiol concentration in comparison with the control group. The results revealed that serum estradiol and testosterone deficiency was induced by bilateral OVX surgery. In which groups was this significant? Stating the statistics in the results section as well as the table is not redundant.
Figure 1 You are still not stating exactly what B-G are, neither in the results section, nor the figure legend. It is not sufficient to just write 1B-G and not explain the differences between the figures.
Why were scale bars only added to certain figures and not all? The text is also way too small to see. It is difficult to see even at 200 magnification.
Materials and methods
You shouldn’t begin a sentence with a number, change to Eight-week-old
Was there a specific volume by weight for saline (ml/kg) or how did you determine how much to inject in that span?
Thank you for adding the schematic image of the experimental procedure, it is very helpful.
Please add in section 4.1 that OVX is described in further detail below when it is mentioned.
Section 4.2. – What percentage of isoflurane? Did they receive any analgesics? I am confused, in this section you wrote that “Sham-operated 814 rats were submitted the same surgical procedure, except the ovaries were not removed.” However, in your response you wrote that “The Control group, MetS group, MetS + L-arginine group, and MetS + L-NAME group did not receive SHAM surgery.”. Who received SHAM surgery? I feel like you need to motivate why all the groups didn’t get surgery, one group went through a major procedure and one did not, this can conflict your results.
In section 4.4 you use BP, is that explained anywhere?
Make sure abbreviations like GOP, GPT, LDL, HDL etc are explained the FIRST time that they are mentioned.
Comments on the Quality of English LanguageEnglish language is fine.
Author Response
Comments and Suggestions for Authors
While the authors have addressed certain comments from the last round they have not addressed my comments regarding the statistical results. In addition, there are some requests were not carried out fully, for example you added the letters for each photo of the liver as requested but you do not state what group each photo is a representative image of. In addition, you added scale bars in certain figures but not others and they are teeny tiny.
Introduction
- Line 81 – dietS not diet.
Response: As suggested by the reviewer, we have changed “Diet” to “Diets”. The changes made in the manuscript is marked in red font (please refer to page 2, line 81).
Results
- Please add the descriptive statisticsfor each group, F-statistic, degrees of freedom, p-value, effect size. When adding p-value in text please add the exact p-value not the definition of the asterix. In addition, you stated that you were analyzing based on diet, OVX and treatment, but you are not indicating anything about what were results were? Were there any interactions between these factors? Was the main effect of OVX, diet or treatment significant? When are you providing us with the results of post hoc analysis, none of this is described.
Response: Thank you for your valuable suggestions.
First, we have now included descriptive statistics for each group, along with F-statistics, degrees of freedom, p-values, and effect sizes (please refer to lines 192-193, 358-360, 407-409, 459-460, 506-509, 543-546, 577-579 in the revised manuscript). We have also added the exact p-values in the text, replacing the previous asterisk notation. For example, in Table I, OVX significantly decreased serum estradiol concentration compared to the control group (F (6, 98) = 59.4048, p < 0.0001). Specifically, the MetS + OVX, MetS + OVX + L-arginine, and MetS + OVX + L-NAME groups showed a significant reduction.
We have provided a detailed analysis of the interactions between diet, OVX, and treatment using a two-way ANOVA. The main effects of OVX, diet, and treatment were all found to be significant. Additionally, significant interactions were identified between these factors. Post hoc analysis (Tukey’s HSD test) revealed that L-arginine treatment significantly improved bladder contractile function compared to other groups. Furthermore, we have expanded the results section to offer specific comparisons between the groups. We now detail the interactions and main effect analyses, focusing on the impact of diet, OVX, and treatment. For instance, for the data presented in Table I, we clearly indicate which groups showed significant differences and provide all relevant statistical details within the text and table for transparency.
Second, both metabolic syndrome (MetS) and postmenopausal status with ovarian hormone deficiency (OHD) are linked to bladder overactivity and lower urinary tract symptoms, as described in previous studies (Maturitas. 2009 Apr 20;62(4):362-5). Our previous research (Scientific Reports 2018; 8: 5358) showed that a high-fat, high-sugar (HFHS) diet for 12 months induced MetS and overactive bladder (OAB) in rats. Furthermore, combining MetS with OVX exacerbated bladder dysfunction more than MetS alone.
Our earlier published findings highlighted that the MetS, MetS + OVX, and MetS + OVX + EGCG groups experienced significant increases in bladder weight, urine glucose, urine protein, and the ratio of urine protein to creatinine compared to the control group (Scientific Reports 2018; 8: 5358). Notably, renal function deteriorated more significantly in the MetS + OVX group. These adverse effects were mitigated by EGCG treatment.
Continuing our experimental research, we aimed to investigate the role of nitric oxide (NO) in MetS-induced bladder dysfunction. The present study primarily focuses on evaluating the effect of NO on bladder dysfunction in rats subjected to an HFHS diet, with or without OVX-induced OAB, in the presence of the NO precursor (L-arginine) and/or the NOS inhibitor (L-NAME). This study provides critical insights into the potential therapeutic role of NO in addressing bladder dysfunction in individuals with MetS and postmenopausal status with OAB.
Reference
- Hyperinsulinaemia, a key factor of the metabolic syndrome in postmenopausal women. Gaspard U. Maturitas. 2009 Apr 20;62(4):362-5.
- Epigallocatechin-3-gallate alleviates bladder overactivity in a rat model with metabolic syndrome and ovarian hormone deficiency through mitochondria apoptosis pathways. Yi-Lun Lee, Kun-Ling Lin, Bin-Nan Wu, Shu-Mien Chuang, Wen-Jeng Wu, Yung-Chin Lee, Wan-Ting Ho, Yung-Shun Juan. Scientific Reports 2018; 8: 5358.
- You need to describe these statistics: Table Ishows OVX significantly decreased serum estradiol concentration in comparison with the control group. The results revealed that serum estradiol and testosterone deficiency was induced by bilateral OVX surgery. In which groups was this significant? Stating the statistics in the results section as well as the table is not redundant.
Response: As suggested by the reviewer, we have expanded the description of the statistical analysis in both the results section and Table 1 to provide clarity and avoid redundancy. The following has been added to the results section (please refer to page 5, lines 177-195):
*"In Table 1, as compared to the control group (44.8 ± 8.5 pg/mL), the serum estradiol concentration was 43.5 ± 7.1 pg/mL for the MetS group, 36.6 ± 5.7 pg/mL for the MetS + L-arginine group, and 38.4 ± 7.4 pg/mL for the MetS + L-NAME group, respectively. There was no significant difference in serum estradiol concentrations among the groups that did not receive OVX treatment. However, four weeks after OVX treatment, the serum estradiol concentration significantly decreased: 18.9 ± 2.1 pg/mL in the MetS + OVX group (p = 0.001), 19.6 ± 3.3 pg/mL in the OVX + MetS + L-arginine group (p = 0.001), and 20.3 ± 3.9 pg/mL in the OVX + MetS + L-NAME group (p = 0.001), all compared to the control group.
Similarly, the serum testosterone concentration also significantly decreased four weeks after OVX treatment: 35.0 ± 4.7 pg/mL in the MetS + OVX group (p = 0.001), 36.4 ± 9.3 pg/mL in the OVX + MetS + L-arginine group (p = 0.001), and 33.2 ± 4.7 pg/mL in the OVX + MetS + L-NAME group (p = 0.001) compared to the control group (61.2 ± 10.9 pg/mL), the MetS group (67.2 ± 13.6 pg/mL), the MetS + L-arginine group (64.8 ± 12.2 pg/mL), and the MetS + L-NAME group (61.4 ± 9.9 pg/mL).*
Table I shows that OVX significantly decreased both serum estradiol (F (6, 98) = 59.4048, p < 0.0001) and testosterone (F (6, 98) = 37.0431, p < 0.0001) concentrations compared to the control and MetS groups without OVX. These results clearly indicate that serum estradiol and testosterone deficiencies were induced by bilateral OVX surgery."*
By providing both the exact p-values and descriptive statistics in the results section and Table 1, we ensure that readers have a clear understanding of the significant differences between groups. This additional information enhances the transparency of the statistical findings and aligns with the reviewer's request.
- Figure 1 You are still not stating exactly what B-G are, neither in the results section, nor the figure legend. It is not sufficient to just write 1B-G and not explain the differences between the figures.
Response: As suggested by the reviewer, we have revised the description in Figure legend on page 9, lines 249-252.
- Why were scale bars only added to certain figures and not all? The text is also way too small to see. It is difficult to see even at 200 magnification.
Response: Thank you for your valuable recommendation. We have now ensured that grey scale bars (100 μm) are included in all figures for consistency and clarity. Additionally, we have adjusted the figure legends to include the specific magnification used, as follows: “magnification X 400. Scale bar (grey) = 100 μm.” This has been applied to Figure 4 (page 16, line 382), Figure 5 (page 19, line 428), Figure 6 (page 22, line 475), and Figure 7 (page 25, line 529).
Furthermore, we have increased the font size in all figures to ensure that the text is legible, even at 200x magnification. These adjustments should greatly enhance the readability and visual clarity of the figures.
Methods
- You shouldn’t begin a sentence with a number, change to Eight-week-old.
Response: Thanks for your recommendation. We have revised "8-week-old female Sprague-Dawley rats" to "Eight-week-old female Sprague-Dawley rats" in the Materials and Methods section on page 34, line 813.
- Was there a specific volume by weight for saline (ml/kg) or how did you determine how much to inject in that span?
Response: Thank you for your question. We have clarified the method used to determine the volume of saline and the dosages of L-arginine and L-NAME for intraperitoneal injections in rats. A fixed volume of 0.9% normal saline (500–700 μL) was used to dilute L-arginine (NO precursor, IP 2 mg/kg) and L-NAME (NO inhibitor, IP 10 mg/kg). The dosage was adjusted according to the rats' body weight.
For example:
- For rats weighing 301-400 g, 500 μL of 0.9% normal saline was used for both the control group and the treatment groups.
- As the rats gained weight (e.g., 401-500 g), the volume of saline was increased to 600 μL, and the corresponding amounts of L-arginine and L-NAME were adjusted to maintain the correct dosage per kilogram.
Specifically:
- In the control group fed a normal rat chow diet, the volume of intraperitoneal saline was adjusted from 500 μL to 600 μL as the rats' body weight increased from 301-500 g.
- In the MetS + L-arginine group fed an HFHS diet, 500 μL of normal saline was combined with 0.8 mg L-arginine for rats weighing 400 g, and 600 μL was combined with 1.0 mg L-arginine for rats weighing 500 g.
- In the MetS + L-NAME group, 500 μL of normal saline was combined with 4.0 mg L-NAME for rats weighing 400 g, and 600 μL was combined with 5.0 mg L-NAME for rats weighing 500 g.
We have revised the description of the dosage in the Materials and Methods section on page 35, lines 831-841.
Below is a table that explains the intraperitoneal dosage:
Body weight |
Normal saline (IP) |
Normal saline + L-arginine (IP, 2 mg/kg) |
Normal saline + L-NAME (IP, 10 mg/kg) |
300-400 g |
500 uL |
500 uL normal saline + 0.8 mg L-arginine |
500 uL normal saline + 4 mg L-arginine |
401-500 g |
600 uL |
600 uL normal saline + 1.0 mg L-arginine |
600 uL normal saline + 5 mg L-arginine |
501-600 g |
700 uL |
700 uL normal saline + 1.2 mg L-arginine |
700 uL normal saline + 6 mg L-arginine |
This adjustment ensures that the dosages of both saline and drugs are consistent with the rats' body weight, providing an accurate treatment across all groups.
- Thank you for adding the schematic image of the experimental procedure, it is very helpful.
Response: Thank you for the compliment.
- Please add in section 4.1 that OVX is described in further detail below when it is mentioned.
Response: As recommended by the reviewer, we have included the description as ' In the text, OVX stands for bilateral ovariectomy. OVX is described in further detail below when it is mentioned' in the section 4.1 on page 35, lines 831-832.
- Section 4.2. – What percentage of isoflurane? Did they receive any analgesics?
Response: Thank you for your inquiry. During the experiment, the concentration of isoflurane used for anesthesia was 2-3%, which was adjusted based on the rats' response to maintain an appropriate level of anesthesia. Postoperatively, if any signs of discomfort or pain were observed, the animals received subcutaneous injections of Ketoprofen, an analgesic, at a dose of 0.2-0.5 mg/100 g body weight.
We have revised the “Materials and Methods” section accordingly (page 35, lines 848 and 851-852) to reflect this information clearly.
- Section 4.2. – I am confused, in this section you wrote that “Sham-operated 814 rats were submitted the same surgical procedure, except the ovaries were not removed.” However, in your response you wrote that “The Control group, MetS group, MetS + L-arginine group, and MetS + L-NAME group did not receive SHAM surgery.”. Who received SHAM surgery? I feel like you need to motivate why all the groups didn’t get surgery, one group went through a major procedure and one did not, this can conflict your results.
Response: I apologize for the confusion regarding the sham-operated rats. To clarify, in the current study, the Control group, MetS group, MetS + L-arginine group, and MetS + L-NAME group did not receive SHAM surgery. We have removed the misleading statement that "Sham-operated rats were submitted to the same surgical procedure, except the ovaries were not removed" from Section 4.2 (page 35, lines 852-853).
Initially, the study design included eight groups: the Control group, MetS group, MetS + L-arginine group, MetS + L-NAME group, Sham surgery group, OVX + MetS group, OVX + MetS + L-arginine group, and OVX + MetS + L-NAME group. However, after 12 months of treatment, our analyses of morphology, bladder function, and protein expression indicated that there were no statistically significant differences between the Control group and the Sham surgery group.
Therefore, we decided not to include the sham group in the final tables and figures. This decision was made to simplify the presentation of results, as the sham group did not provide additional insights beyond what was observed in the control group.
Thank you for your insightful recommendation, and we hope this explanation clarifies the rationale behind the exclusion of the sham group from the final analysis.
- In section 4.4 you use BP, is that explained anywhere?
Response: Thank you for your suggestion. We have revised abbreviation “BP” to “blood pressure (BP) “ in section 4.4 (please refer to page 36, line 872).
- Make sure abbreviations like GOP, GPT, LDL, HDL etc are explained the FIRST time that they are mentioned.
Response: Thank you for your suggestion. We have carefully reviewed the document and ensured that all abbreviations, such as GOP, GPT, LDL, HDL, and others, are explained the first time they are mentioned. Specific revisions have been made in the following sections to clarify these abbreviations: page 4, line 141, 155-156; page 8, line 229-232; page 10, line 266; page 12, line 313-314; page 19, line 440; page 29, line 627).
Comments on the Quality of English Language
- English language is fine.
Response: Thank you for your positive feedback regarding the quality of the English language.
